# Integrating Large Language Models in Causal Discovery: A Statistical Causal Approach

## Abstract

In practical statistical causal discovery (SCD), embedding domain expert knowledge as constraints into the algorithm is significant for creating consistent meaningful causal models, despite the challenges in systematic acquisition of the background knowledge. To overcome these challenges, this paper proposes a novel method for causal inference, in which SCD and knowledge based causal inference (KBCI) with a large language model (LLM) are synthesized through "statistical causal prompting (SCP)" for LLMs and prior knowledge augmentation for SCD. Experiments have revealed that the results of LLM-KBCI and SCD augmented with LLM-KBCI approach the ground truths, more than the SCD result without prior knowledge. It has also been revealed that the SCD result can be further improved if the LLM undergoes SCP. Furthermore, with an unpublished real-world dataset, we have demonstrated that the background knowledge provided by the LLM can improve SCD on this dataset, even if this dataset has never been included in the training data of the LLM. The proposed approach can thus address challenges such as dataset biases and limitations, illustrating the potential of LLMs to improve data-driven causal inference across diverse scientific domains.

## 1 Introduction

### 1.1 Background

Understanding causal relationships is key to comprehending basic mechanisms in various scientific fields. The statistical causal inference framework, which is widely applied in areas such as medical science, economics, and environmental science, aids this understanding. However, traditional statistical causal inference methods generally rely on the assumed causal graph for determining the existence and strength of causal impacts. To overcome this challenge, data-driven algorithmic methods have been developed as statistical causal discovery (SCD) methods, both in non-parametric (Spirtes et al., 2000; Chickering, 2002; Silander & Myllymäki, 2006; Yuan & Malone, 2013; Huang et al., 2018; Xie et al., 2020) and semi-parametric (Shimizu et al., 2006; Hoyer et al., 2008; Shimizu et al., 2011; Rolland et al., 2022; Tu et al., 2022) approaches. In addition, benchmark datasets have been published for the evaluation of SCD methods (Mooij et al., 2016; Käding & Runge, 2023).

Despite advancements in SCD algorithms, data-driven acquisition of causal graphs without domain knowledge can be inaccurate. This is generally attributed to a mismatch between assumptions in SCD and real-world phenomena (Reisach et al., 2021). Moreover, obtaining experimental and systematic datasets sufficient for causal inference is difficult, whereas observational datasets, which are prone to selection bias and measurement errors, are more readily accessible (Abdullahi et al., 2020). Consequently, for more persuasive and reliable validation of causal models, the augmentation with domain knowledge plays a critical role (Rohrer, 2017).

In addition, with respect to efficiency and precision in SCD, the importance of incorporating constraints on trivial causal relationships into the SCD algorithms has been highlighted (Inazumi et al., 2010; Chowdhury et al., 2023). Causal learning software packages have been augmented with prior knowledge, as demonstrated in "causal-learn" [1], and "LiNGAM" [2] (Zheng et al., 2023; Ikeuchi et al., 2023).

---

[1] https://github.com/py-why/causal-learn
[2] https://github.com/cdt15/lingam

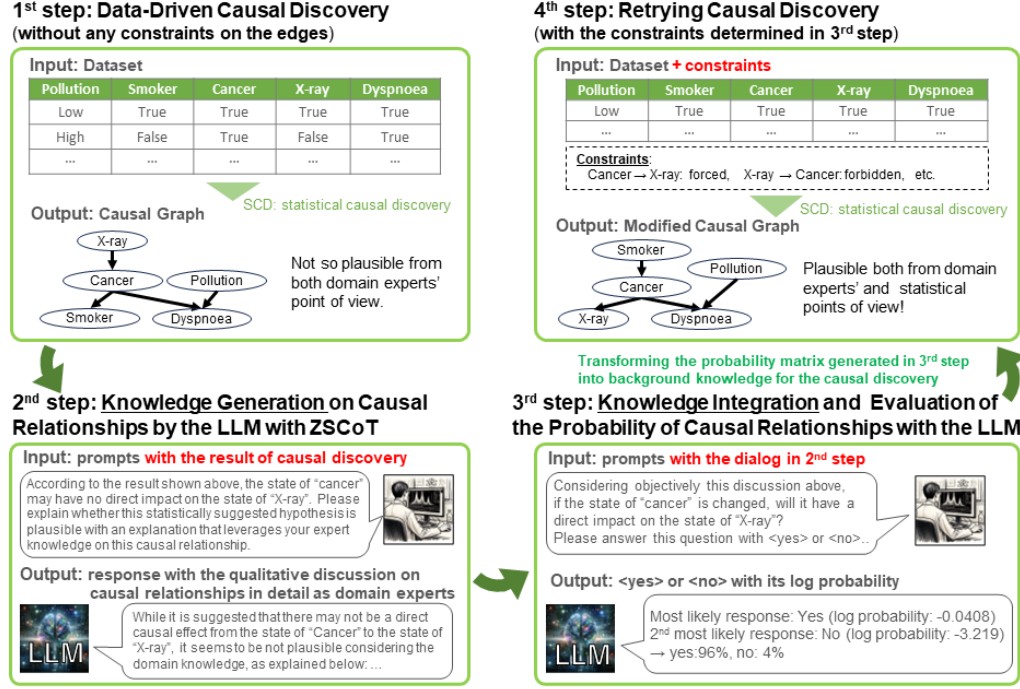

Figure 1: Overall framework of the statistical causal prompting in a large language model (LLM) and statistical causal discovery (SCD) with LLM-based background knowledge.

Moreover, the systematic acquisition of domain expert knowledge is a challenging task. Although there are several examples of constructing directed acyclic graphs (DAGs) by domain experts, as demonstrated in health services research (Rodrigues et al., 2022), practical methods for this process have not been proposed.

The scenario recently has been changed with the rapid progress in the development of high-end large language models (LLMs). With their high performances in the applications of their domain knowledge acquired from vast amounts of data in the pre-training processes (OpenAI, 2023; Touvron et al., 2023; Gemini Team, Google, 2023), it has been expected that LLMs also can be applied to the causal reasoning tasks. In fact, several studies have reported the trial results of LLM knowledge-based causal inference (LLM-KBCI) (Jiang et al., 2023; Jin et al., 2024; Kıcıman et al., 2023; Zečević et al., 2023; Jiralerspong et al., 2024; Zhang et al., 2024), and in particular, the performance enhancement in non-parametric SCD with the guides by LLMs was confirmed (Ban et al., 2023; Vashishtha et al., 2023; Khatibi et al., 2024). However, it remains unclear whether the enhancement in SCD accuracy with background knowledge augmented by LLMs is robustly observed when the task of the SCD depends on closed data uncontained in the pre-training datasets of LLMs, and whether it leads to more statistically valid causal models.

## 1.2 Central Idea of Our Research

Based on the rapidly evolving techniques in the context of causal inference with LLMs, a novel methodology for SCD is proposed in this paper, in which the LLM prompted with the results of SCD without background knowledge evaluates the probability of the causal relationships considering both the domain knowledge and the statistical characteristic suggested by SCD (Figure 1).

In the first step, an SCD is executed on a dataset without prior knowledge, and the results of the statistical causal analysis are outputted. To maximize the usage of the expert knowledge acquired in the pre-training process of the LLM, the method of generated knowledge prompting (Liu et al., 2022) is adopted, and then, the process of utilizing the LLM includes the second step (knowledge generation) and the third step (knowledge

integration). In the second step, knowledge of the causal relationships between all pairs of variables is generated in detail from the domain knowledge of the LLM based on the zero-shot chain-of-thought (ZSCoT) prompting technique (Kojima et al., 2022). Here, the LLM can be prompted with the results of SCD as supplemental information for LLM inference, expecting that the in-context learning (Brown et al., 2020) of the SCD results can leads to the better performance of the LLM-KBCI in terms of statistic. We define this technique as "statistical causal prompting" (SCP). Thereafter, in the third step, the LLM judges whether there are any causal relationships between all pairs of variables with "yes" or "no," thus objectively considering the dialogs of the second step. Here, the probabilities of the responses from the LLM are evaluated and transformed into the prior knowledge matrix. This matrix, the output of LLM-KBCI with SCP, is finally re-applied to SCD in the fourth step.

### 1.3 Our Contribution

The contributions through the demonstration of the proposed method in this paper are as follows:

**(1) Realization of the Synthesis of LLM-KBCI and SCD in a Mutually-Referential Manner**
The practical method for realizing the proposed concept of Figure 1 is detailed, and the SCP method is proposed. Experiments were demonstrated with several benchmark datasets, which are open and have been widely used for the evaluation of SCD algorithms. They all consist of continuous variables.

**(2) Mutual Performance Enhancement of SCD and LLM-KBCI** We demonstrated that the augmentation by the LLM with SCP, improved the performance of the SCD, and the performance of LLM-KBCI was also enhanced by SCP.

**(3) Enhancement of SCD Performance by SCP** In the experiments, we demonstrate the implication that the output of several SCD algorithms augmented by the LLM with SCP, can be a superior causal model to the pattern of prompting without SCP in terms of domain expertise and statistic.

**(4) Improvement of SCD results with the background knowledge provided by LLMs even if the dataset is uncontained in the pre-training materials** We prepared a closed health screening dataset that is uncontained in the pre-training materials for LLMs, and demonstrated the experiment on a sub-dataset that has been randomly sampled from the entire dataset. Through this experiment, we clearly confirmed that the proposed method robustly leads to more statistically valid causal models with the natural ground truths. This fact proved that the LLMs can indeed contribute to background knowledge augmentation for SCD algorithms in practical situations, even if the dataset used for SCD are not memorized by LLMs.

## 2 Related Works and Originality in Our Work

**Augmentation of SCD Algorithms with Background Knowledge** As introduced in Section 1.1, several SCD algorithms [3] can be systematically augmented with background knowledge. Moreover, their software packages are open. For example, as a non-parametric and constraint-based SCD method, the Peter-Clerk (PC) algorithm (Spirtes et al., 2000) is augmented with the background knowledge of the forced or forbidden directed edges in "causal-learn." "Causal-learn" also provides the Exact Search algorithm (Silander & Myllymäki, 2006; Yuan & Malone, 2013) as a non-parametric and score-based SCD method, which can be augmented with the background knowledge of the forbidden directed edges as a super-structure matrix. With respect to a semi-parametric approach, DirectLiNGAM (Shimizu et al., 2011) algorithm is augmented with prior knowledge of the causal order (Inazumi et al., 2010) in "LiNGAM" project (Ikeuchi et al., 2023).

**Causal Inference in Knowledge-Driven Approach with LLMs** In addition to the study on the causality detection from natural language texts using language models with additional training datasets (Khetan et al., 2022), the rapid growth of LLMs have made it possible to produce some valuable works on

---

[3]All of the algorithms adopted for the experiments in this paper can be used under the assumptions of DAG and with no hidden confounders.

causality, including causal inference using LLMs. Attempts have been made to use LLMs for causal inference among a set of variables, prompting with the metadata such as the names of variables, and without the SCD process with the benchmark datasets (Kıcıman et al., 2023; Zečević et al., 2023; Jiralerspong et al., 2024; Zhang et al., 2024). Adopting the similar approach, the concept of causal modeling agents, which improves the precision of the causal graphs through the iteration of hypothesis generation from LLMs and model fitting to the real-world data, has also been proposed (Abdulaal et al., 2024). There are also studies on incorporating LLMs in the process of SCD as an alternative tool for conditional independence tests in the PC algorithm (Cohrs et al., 2023), and for identifying the causal graphs beyond the Markov equivalent class (Long et al., 2023). In addition, researches have been conducted with focus on the use of LLMs to improve the SCD results(Ban et al., 2023; Vashishtha et al., 2023; Khatibi et al., 2024). However, all the experiments were conducted only on popular benchmark datasets, contained in the pre-training datasets of LLMs. Consequently, while acknowledging the valuable foundations laid by previous studies, it has remained uncertain whether the enhancements in SCD accuracy are truly driven by LLMs leveraging their vast knowledge for genuine causal inference, or merely by reproducing the memorized content of datasets (Vashishtha et al., 2023)..

**Originality in Our Work**  In contrast to the studies with similar focuses on LLM-guided SCD (Ban et al., 2023; Vashishtha et al., 2023; Khatibi et al., 2024), this study also focuses on the construction of the background knowledge in a quantitative manner based on the response probability of the LLM, which can reflect the credibility of the decision made by the LLM with SCP. In addition, in the case of a semi-parametric SCD method such as LiNGAM, we detail herein how to achieve both statistical validity and natural interpretation with respect to domain knowledge at a maximally high level, by prompting with the causal coefficients and bootstrap probabilities of all the patterns of directed edges.

Moreover, we validate the proposed method on an unpublished dataset. This approach has not only demonstrated the practical utility and robustness of our method, but also opened new avenues for supporting the validity and applicability of existing works (Ban et al., 2023; Vashishtha et al., 2023; Khatibi et al., 2024).

## 3  Materials and Methods

### 3.1  Algorithms and Elements for LLM-Augmented Causal Discovery

With respect to practicality, the method of Figure 1 is outlined as Algorithm 1. Following the notation in Algorithm 1, the input elements in the demonstration are explained below.

**Algorithms for Statistical Causal Discovery**  For the SCD method $S$, we adopted the PC algorithm (Spirtes et al., 2000), Exact Search based on the A$^*$ algorithm (Yuan & Malone, 2013), and DirectLiNGAM algorithm (Shimizu et al., 2011), which can all be optionally augmented with prior knowledge, and are open in "causal-learn" (Zheng et al., 2023) and "LiNGAM" (Ikeuchi et al., 2023). Furthermore, we also implement the bootstrap sampling function $B$ of the SCD algorithm to investigate the statistical properties such as the bootstrap probabilities $p_{ij}$ of the emergence of the directed edges from $x_j$ to $x_i$. In our experiments, the number of bootstrap resamplings was fixed to 1000.

**Conditions of the LLM and Prompting**  For utilizing the LLM as the domain expert, we adopted `GPT-4-1106-preview`[4] developed by OpenAI; the temperature, a hyperparameter for adjusting the probability distribution of the output, was fixed to 0.7.

---

[4]We recognize that there have been various kinds of high-performance LLMs from several institutes, and that it is valuable to demonstrate that including a broader range of LLMs could provide valuable insights into the generalizability and scalability of our method across various LLM architectures. However, our goal in this work is to explore the effectiveness of integrating LLMs into SCD via SCP, requiring trials and comparisons of various SCP patterns and SCD algorithms, as described in Section 4. To maintain consistency and control across these trials, it is also important to fix the LLM in the experiments, which has advanced capabilities and state-of-the-art status in various domains. Moreover, we should adopt the LLMs that satisfy specific conditions for our experiments, such as the maximum input token capacity for the prompting processes, and the functionality to obtain log-probability of the output. For these strategic and technical reasons, we adopted GPT-4 in our experiments.

---

**Algorithm 1** Background knowledge construction with the LLM prompted with the results of the SCD

---

**Input 1:** Data $X$ with variables $\{x_1, ..., x_n\}$
**Input 2:** SCD method $S$
**Input 3:** Function for bootstrap $B$
**Input 4:** Response of the domain expert (GPT-4) $\epsilon_{\text{GPT4}}$
**Input 5:** Log probability of the response $L(\epsilon_{\text{GPT4}})$
**Input 6:** Prompt function for knowledge generation $Q_{ij}^{(1)}$
**Input 7:** Prompt function for knowledge integration $Q_{ij}^{(2)}$
**Input 8:** Transformation from probability matrix to prior knowledge $T$
**Input 9:** Number of times to measure the probability $M$
**Output:** Result of SCD with prior knowledge $\hat{G}$ on $X$

SCD result without prior knowledge $\hat{G}_0 = S(X)$
bootstrap probability matrix $\boldsymbol{P} = B(S, X)$
**for** $i = 1$ **to** $n$ **do**
   **for** $j = 1$ **to** $n$ **do**
      $\overline{p_{i,i}} = \text{NaN}$
      **if** $i \neq j$ **then**
         prompt $q_{ij}^{(1)} = Q_{ij}^{(1)}(x_i, x_j, \hat{G}_0, \boldsymbol{P})$
         response $a_{ij} = \epsilon_{\text{GPT4}}(q_{ij}^{(1)})$
         prompt $q_{ij}^{(2)} = Q_{ij}^{(2)}(q_{ij}^{(1)}, a_{ij})$
         **for** $m = 1$ **to** $M$ **do**
            $L_{ij}^{(m)} = L^{(m)}(\epsilon_{\text{GPT4}}(q_{ij}^{(2)}) = \text{"yes"})$
         **end for**
         mean probability $\overline{p_{ij}} = \dfrac{\sum_{m=1}^{M} \exp(L_{ij}^{(m)})}{M}$
      **end if**
   **end for**
**end for**
probability matrix $\overline{\boldsymbol{p}} = (\overline{p_{ij}})$
prior knowledge $\boldsymbol{PK} = T(\overline{\boldsymbol{p}})$
**return** $\hat{G} = S(X, \boldsymbol{PK})$

---

The template for the first prompting $q_{ij}^{(1)}$ for knowledge generation is shown in Table 1. This prompting template is based on the underlying principle of the ZSCoT technique[5] (Kojima et al., 2022), which was reported as a potential method to enhance the performance of the LLM generation tasks; enhancement is performed by guiding logical inference and eliciting the background knowledge acquired through the pre-training process from the LLM. Furthermore, expecting that the in-context learning (Brown et al., 2020) of the SCD results can leads to the better performance of the LLM-KBCI in terms of statistic, the SCD results, e.g., the causal structure and bootstrap probabilities, can be included in ⟨blank 5⟩ and ⟨blank 9⟩, which are defined as "statistical causal prompting" (SCP). Because the information used in SCP is partially dependent on the SCD algorithms, a brief description of the patterns for constructing the contents of ⟨blank 5⟩ and ⟨blank 9⟩ is presented in Section 3.2.

As shown in Table 2, generated knowledge is integrated in the second prompt, and GPT-4 is required to judge the existence of the causal effect from $x_j$ on $x_i$ from an objective point of view. Because the response from GPT-4 is required with "yes" or "no," it is simple to quantitatively evaluate the level of GPT-4's confidence regarding the existence of the causal relationship based on both the SCD result and domain knowledge. The probability $p_{ij}$ of the assertion that there is a causal effect from $x_j$ on $x_i$ can be output from GPT-4

---

[5]Although the quality of the LLM outputs can be further enhanced, e.g., by fine-tuning with several datasets containing fundamental knowledge for causal inference or Retrieval-Augmented Generation (RAG), we adopt the idea of ZSCoT in order to establish low-cost and simple methods, which can be universally applied independent of the targeted fields of causal inference.

Table 1: Prompt template of $Q_{ij}^{(1)}(x_i, x_j, \hat{G}_0, \boldsymbol{P})$ used for the generation of the expert knowledge of the causal effect from $x_j$ on $x_i$. The "blanks" enclosed with $\langle \rangle$ are filled with description words . The word for $\langle$blank 6$\rangle$ is selected from "a" or "no," depending on the content of $\langle$blank 5$\rangle$. The notations of the SCD result without prior knowledge $\hat{G}_0$ and the bootstrap probability matrix $\boldsymbol{P}$ are same as those in Algorithm 1.

---

**Prompt Template of $q_{ij}^{(1)} = Q_{ij}^{(1)}(x_i, x_j, \hat{G}_0, \boldsymbol{P})$**

---

We want to carry out causal inference on $\langle$blank 1. The theme$\rangle$, considering $\langle$blank 2. The description of all variables$\rangle$ as variables.
First, we have conducted the statistical causal discovery with $\langle$blank 3. The name of the SCD algorithm$\rangle$, using a fully standardized dataset on $\langle$blank 4. The description of the dataset$\rangle$.

$\langle$blank 5. Including here the information of $\hat{G}_0$ or $\boldsymbol{P}$.The detail of the contents depends on prompting patterns.$\rangle$

According to the results shown above, it has been determined that there may be $\langle$blank 6. a/no$\rangle$ direct impact of a change in $\langle$blank 7. The name of $x_j\rangle$ on $\langle$blank 8. The name of $x_i\rangle$ $\langle$blank 9. The value of causal coefficients or bootstrap probability$\rangle$.
Then, your task is to interpret this result from a domain knowledge perspective and determine whether this statistically suggested hypothesis is plausible in the context of the domain.

Please provide an explanation that leverages your expert knowledge on the causal relationship between $\langle$blank 7. The name of $x_j\rangle$ and $\langle$blank 8. The name of $x_i\rangle$, and assess the naturalness of this causal discovery result. Your response should consider the relevant factors and provide a reasoned explanation based on your understanding of the domain.

---

directly with the optional function of returning log-probabilities of most likely tokens at each token position[6]. Although $p_{ij}$ can be evaluated readily from the log probability of the GPT-4 response as "yes," there is a slight fluctuation in the log probability output from GPT-4 (Andriushchenko et al., 2024). Thus, we adopted the mean probability $\overline{p_{ij}}$ of the single-shot measurement $M$ times for the decision of prior knowledge matrix $\boldsymbol{PK}$, and we set $M = 5$ in the experiments. The combination of these prompting techniques can contribute to minimizing the risk of hallucination from the LLM, and it is expected that reliable probability of the response from the LLM for the causal relationship between a pair of variables is obtained.

Table 2: Prompt template of $Q_{ij}^{(2)}(q_{ij}^{(1)}, a_{ij})$ used for the quantitative evaluation of the probability of GPT-4's assertion that there is a causal effect from $x_j$ on $x_i$.

---

**Prompt Template of $q_{ij}^{(2)} = Q_{ij}^{(2)}(q_{ij}^{(1)}, a_{ij})$**

---

An expert was asked the question below:
$\langle$blank 10. $q_{ij}^{(1)}\rangle$

Then, the expert replied with its domain knowledge:
$\langle$blank 11. $a_{ij}\rangle$

Considering objectively this discussion above, if $\langle$blank 12. The name of $x_j\rangle$ is modified, will it have a direct or indirect impact on $\langle$blank 13. The name of $x_i\rangle$?
Please answer this question with $\langle$yes$\rangle$ or $\langle$no$\rangle$.
No answers except these two responses are needed.

---

For the subsequent SCD with prior knowledge, the probability matrix $\overline{\boldsymbol{p}}$ is transformed with $T$, as expressed by Algorithm 2, into the background knowledge matrix. Here, if $PK_{ij} =$ Forbidden, the causal effect from $x_j$ to $x_i$ is forbidden to appear in the SCD result. On the other hand, if $PK_{ij} =$ Forced, the causal effect from

---

[6]The technical detail of this probability sampling is explained in Appendix D .

---

**Algorithm 2** Transformation from the probability matrix into the prior knowledge matrix

---
**Input 1:** probability matrix $\boldsymbol{p} = (p_{ij})$
**Input 2:** SCD method $S \in \{$ PC, Exact Search, DirectLiNGAM $\}$
**Input 3:** probability criterion for the forbidden causal relationship $\alpha_1$
**Input 4:** probability criterion for the forced causal relationship $\alpha_2$
**Output:** prior knowledge matrix $\boldsymbol{PK} = (PK_{ij})$
**for** $i = 1$ **to** $n$ **do**
  **for** $j = 1$ **to** $n$ **do**
    $PK_{ij} = $ Unknown
    **if** $i = j$ **then**
      $PK_{ij} = $ Forbidden
    **else**
      **if** $p_{ij} < \alpha_1$ **then**
        $PK_{ij} = $ Forbidden
      **else if** $(p_{ij} \geq \alpha_2)$ **and** $(S \neq$ Exact Search$)$ **then**
        $PK_{ij} = $ Forced
      **end if**
    **end if**
  **end for**
**end for**
**if** $S = $ DirectLiNGAM **then**
  $\boldsymbol{PK} = A(\boldsymbol{PK})$ (acyclic transformation)
**end if**
**return** $\boldsymbol{PK}$

---

$x_j$ to $x_i$ is forced to appear in the SCD result[7]. For the decision of forbidden or forced causal relationship from $PK_{ij}$, we prepare the probability criterion for the forbidden path as $\alpha_1$ and that for the forced path as $\alpha_2$. In our experiments, $\alpha_1$ is fixed at 0.05 and $\alpha_2$ is fixed at 0.95, for common settings[8].

Furthermore, the differences in the constraints that can be adopted depending on the SCD algorithms should be considered. In the Exact Search algorithm, the constraints of the forced edge cannot be applied. For the case of DirectLiNGAM, because prior knowledge is used for the decision of the causal order in the algorithm, the prior knowledge matrix must be an "acyclic" adjacency matrix when it is represented in the form of a network graph. Thus, when $S = $ DirectLiNGAM and $\boldsymbol{PK}$ is cyclic, an additional transformation algorithm $A$ is required. In addition, there can be several acyclic transformation patterns; only one acyclic matrix with some criteria should be selected. The algorithm for the transformation and the matrix selection criterion in this study are explained in Appendix F.

### 3.2 Experiment Patterns of SCP

Related to the ⟨blank 5⟩ and ⟨blank 9⟩ in Table 1, we conducted experiments using several patterns of SCP. the notations of the prompting patterns in the experiments are presented with explanations below:

**Pattern 0: without SCP** This pattern corresponds to the reference for the comparison with the other patterns including SCD results in their prompts. Because the prompt template shown in Table 1 is not adequate for this pattern, we prepare a different template for Pattern 0, which is shown in Appendix B.

---

[7]Although these constraints can be systematically adapted with the augmentation supported in "causal-learn" (Zheng et al., 2023) and "LiNGAM" (Ikeuchi et al., 2023), the detailed meanings of forced and forbidden causal effects are slightly different among the SCD algorithms. The details are described in Appendix E

[8]These heuristic thresholds follow the widely accepted conventions seen in statistical significance levels. Although the specific choice of threshold might influence the formation of the prior knowledge matrix, the probability outputs from LLMs for clear domain knowledge instances are either very high (close to 100 %) or very low (closed to 0 %), ensuring that the essential insights are retained regardless of the threshold. In Appendix D, technical details of this sensitivity around these thresholds are described through the evaluation of the standard error of $\overline{p_{ij}}$, which can be realized by single-shot measurements of $\overline{p_{ij}}$ for $M$ times.

**Pattern 1: with the list of edges that appeared in the first SCD** Directed or undirected[9] edges between $x_i$ and $x_j$ emerged in the SCD are listed.

**Pattern 2: with the list of edges with their non-zero bootstrap probabilities** Directed or undirected edges between $x_i$ and $x_j$ that emerged in the bootstrap process are listed with their bootstrap probabilities.

**Pattern 3: with the list of edges that emerged in the first SCD with the calculated causal coefficients (only for DirectLiNGAM)** Based on the property of DirectLiNGAM, that outputs the causal coefficients with the DAG discovered in the algorithm, this pattern is attempted to elucidate whether more information such as causal coefficients in addition to Pattern 1 can improve the performance of LLM-KBCI and the subsequent SCD with prior knowledge.

**Pattern 4: with the list of edges with their non-zero bootstrap probabilities and calculated causal coefficients for the full dataset (only for DirectLiNGAM)** We also attempt this pattern with the most amount of information of 1st SCD as a mixture of Patterns 2 and 3.

### 3.3 Datasets for the Experiments

Although there are several widely-open benchmark datasets with well-known ground truths, particularly for Bayesian network-based causal structure learning (Scutari & Denis, 2014), several of them are fully or partially composed with categorical or discrete variables. However, considering Patterns 3 and 4 for the experiments in this study, because the basic structure causal model assumes the continuous properties of all variables, it is more effective to adopt benchmark datasets fully composed with continuous variables.

Consequently, we select three benchmark datasets for the experiments, as follows: 1. Auto MPG data (Quinlan, 1993) (five continuous variables) , 2. Deutscher Wetterdienst (DWD) climate data (Mooij et al., 2016) (six continuous variables) , 3. Sachs protein data (Sachs et al., 2005)(eleven continuous variables) .

Furthermore, to demonstrate that GPT-4 can aid SCD with its domain knowledge, even if the dataset used in the SCD process and analytics on the dataset are not contained in the pre-training data of GPT-4, the proposed methods are also applied on our dataset on health screening results, which has not been disclosed and therefore not learned by GPT-4. To demonstrate that the proposed methods can be applied when the dataset contains bias, which may lead to highly inaccurate SCD results, the health screening dataset for this experiment was sampled, and we deliberately chose a subset where certain biases are still present. Basic information on these datasets such as the first SCD results and the ground truths is presented in Appendix C.

## 4 Results and Discussions

### 4.1 Results in Benchmark Datasets

For the interpretation of the experimental results, we evaluate $PK$ (for measuring the performance of LLM-KBCI with the prompts, including the first SCD results) and the adjacency matrix obtained in SCD with $PK$ for each pattern (for measuring the performance of SCD augmented with LLM-KBCI), with the structural hamming distance (SHD), false positive rate (FPR), false negative rate (FNR), precision, and F1 score, using the ground truth adjacency matrix $GT$ as a reference. All of these metrics are frequently used to evaluate how the results of the causal structure learning are close to the ground truths, and can be calculated solely from the adjacency matrix and $GT$, as shown in Appendix E.3. In addition, this paper presents the evaluation of the comparative fit index (CFI) (Bentler, 1990), root mean square error of approximation (RMSEA) (Steiger & Lind, 1980) and Bayes information criterion (BIC) (Schwarz, 1978) of the causal structure obtained in SCD with $PK$, under the assumption of linear-Gaussian data[10]; to evaluate the results with respect to the

---

[9]In the PC algorithm, undirected edges that appear as $x_i - x_j$ with respect to a causal relationships between "$x_i$ and $x_j$" in which the direction cannot be determined, can be detected. The prompt template for reflecting this difference from directed edges is shown in Appendix B.

[10]Although this assumption of linear-Gaussian data for the calculation of the CFI, RMSEA, and BIC, does not match the assumption of a non-Gaussian error distribution in LiNGAM, we adopt these indices to evaluate and compare the results with respect to the same statistical method, irrespective of the difference in the SCD algorithms.

statistical validity of calculated causal models. For evaluating the effect of **PK** augmentation by the LLM, the baseline result is that without **PK** (Baseline A), as a reference for the comparison with the SCD results augmented with **PK**. In contrast, for evaluating the effect of SCP, the results with the prompting in Pattern 0 becomes the baseline (Baseline B), in the comparison with the results obtained in other SCP patterns. The indices for all patterns on the DWD, Auto MPG, and Sachs datasets are summarized in Table 3.

**Enhancing the performance with prior knowledge augmentation by GPT-4**   One of the characteristics in Table 3 is that in most of the cases, the result of SCD augmented with **PK** is more similar to **GT** than the first SCD result without prior knowledge (Baseline A). This behavior is interpreted as the knowledge-based improvement of the causal graph by GPT-4 as a domain expert, which is qualitatively consistent with other related works on LLM-guided causal inference (Ban et al., 2023; Vashishtha et al., 2023). Moreover, in many of the cases of Auto MPG and DWD data, the precision or F1 score are higher after the SCD augmented with **PK** than the pure **PK**, which are conclusions of LLM-KBCI, while they are almost comparable in the cases of Sachs data. From this comparison it is implied that even if LLM-KBCI is not optimal, the ground truths can be better approached by conducting SCD augmented with LLM-KBCI. In addition, BIC decreases in almost all the patterns from the SCD result without **PK** (Baseline A). The aforementioned properties suggest that knowledge-based augmentation from GPT-4 can improve the performance of SCD, indeed, in terms of the consistency with respect to the domain expert knowledge and statistical causal structure. However, the amount of improvement can differ depending on the number of variables and the methods of SCD.

**Dependence on the number of variables**   In the case of Auto MPG data with only five variables, the amount of improvement for each SCD method is almost constant among all the prompting patterns. One of the possible reasons is that within relatively small numbers of variables, the amount of information in SCP becomes small, and the difference of inference performance of GPT-4 among the prompting patterns becomes subtle. Moreover, because the space for discovery becomes also smaller along with the network shrink, the SCD algorithm may reach a single optimal solution, even if **PK** is different.

On the other hand, in the cases of DWD data with six variables and Sachs data with eleven variables, the difference in the amount of improvement becomes more clear depending on the prompting patterns. From this fact, it is implied that the threshold of the number of variables over which the quality of **PK** and SCD results depend on the amount of information included. In SCP for GPT-4, it is around five.

Moreover, in many cases of DWD and Sachs data, in particular for Exact Search or DirectLiNGAM, the precision and F1 score of **PK** in Pattern 0 (Baseline B) are usually smaller than any of other patterns, in which GPT-4 experiences SCP. This supports the performance enhancement of LLM-KBCI by SCP.

**Prompting pattern dependence on SCD methods**   On the other hand, considering the output of the SCD augmented with **PK**, Pattern 0 (Baseline B) stably indicates relatively higher performance among all the patterns in both terms of domain knowledge and statistical model fitting. Furthermore, the results of Patterns 0 and 1 are almost the same when the PC or the Exact Search algorithm is adopted, and in particular, the result of Pattern 0 with the PC algorithm on Sachs data is superior to that of Pattern 0.

Although it is difficult to explain the reason for this behavior, one of the possible reasons, if we focus only on the results on Sachs data, is that we adopted the ground truths partially determined by Bayesian network inference (Sachs et al., 2005). Indeed, the first SCD result in PC is already relatively close to **GT**, as shown in Figure 8 (a), and we interpret that in this situation, the performance of SCD with **PK** cannot be improved.

The scenario differs when DirectLiNGAM is adopted. The performance of the SCD with **PK** either in Pattern 1 or 2, remains totally superior to that of Pattern 0 (Baseline B) from both statistical and domain expert points of view. This implies that SCP can effectively improve the performance of DirectLiNGAM.

However, from the analysis of Patterns 3 and 4, in which GPT-4 is prompted with the causal coefficients of the first SCD results, it is also revealed that prompting with a greater amount of statistical information does not always lead to improved SCD results. In particular, while **PK** in Pattern 4 is closer to ground truth matrix than that in Pattern 0 in DWD or Sachs data [11], the final SCD result augmented with **PK** in Pattern

---

[11]This fact reinforces the reliability of our interpretation that SCP enhance the performance of LLM-KBCI.

Table 3: Comparison of the SCD results in all the experiment patterns we have conducted. While SHD, FPR, FNR, precision, and F1score are compared to evaluate how the results are close to the ground truths, CFI, RMSEA, and BIC are compared to evaluate the statistical validity of the calculated causal models. Lower values are superior for SHD, FPR, FNR, RMSEA and BIC, and higher values for precision, F1score and CFI. The numbers in parentheses indicate SHD, FPR, FNR, precision, and F1score of **PK** with ground truths, for the evaluation of the performance of LLM-KBCI. The values in the blue font are the optimal results among Patterns 0–4, If the dataset and the SCD algorithm is fixed. Baseline A is used for the comparison with the SCD results augmented with *PK* generated by the LLM, to evaluate the effect of **PK** augmentation by the LLM. Baseline B is used for the comparison with the results obtained in other SCP patterns, to evaluate the effect of SCP. It is implied that in DWD and Sachs datasets, the outputs of LLM-KBCI in Patterns 1–4 are likely to approach the ground truths more closely than those in Pattern 0. It is also suggested that several of the outputs of Exact Search and DirectLiNGAM in Patterns 1–4 (with SCP), can be superior causal models than those in Pattern 0.

| SCD algorithm | Pattern | SHD↓ | FPR↓ | FNR↓ | Precision↑ | F1score↑ | CFI↑ | RMSEA↓ | BIC↓ |
|---|---|---|---|---|---|---|---|---|---|
| **1. Auto MPG data with 5 continuous variables** | | | | | | | | | |
| PC | wo **PK**(Baseline A) | 8 | 0.40 | 0.80 | 0.11 | 0.14 | 1.00 | 0.00 | 71.65 |
| | Pattern 0 (Baseline B) | 3 (5) | 0.15 (0.25) | 0.20 (0.20) | 0.57 (0.44) | 0.67 (0.57) | 1.00 | 0.07 | 65.62 |
| | Pattern 1 | 4 (7) | 0.15 (0.30) | 0.20 (0.20) | 0.57 (0.40) | 0.67 (0.53) | 1.00 | 0.00 | 59.71 |
| | Pattern 2 | 3 (6) | 0.15 (0.30) | 0.20 (0.20) | 0.57 (0.40) | 0.67 (0.53) | 1.00 | 0.07 | 65.62 |
| Exact Search | wo **PK**(Baseline A) | 5 | 0.25 | 0.40 | 0.38 | 0.46 | 1.00 | 0.07 | 71.61 |
| | Pattern 0 (Baseline B) | 4 (5) | 0.20 (0.25) | 0.20 (0.20) | 0.50 (0.44) | 0.62 (0.57) | 1.00 | 0.09 | 71.59 |
| | Pattern 1 | 5 (5) | 0.20 (0.20) | 0.20 (0.20) | 0.50 (0.50) | 0.62 (0.62) | 1.00 | 0.07 | 65.62 |
| | Pattern 2 | 4 (5) | 0.20 (0.25) | 0.20 (0.20) | 0.50 (0.44) | 0.62 (0.57) | 1.00 | 0.09 | 71.59 |
| DirectLiNGAM | wo **PK**(Baseline A) | 8 | 0.40 | 0.80 | 0.11 | 0.14 | 1.00 | 0.05 | 77.61 |
| | Pattern 0 (Baseline B) | 3 (5) | 0.15 (0.25) | 0.20 (0.20) | 0.57 (0.44) | 0.67 (0.57) | 1.00 | 0.07 | 65.62 |
| | Pattern 1 | 3 (5) | 0.15 (0.30) | 0.20 (0.20) | 0.57 (0.40) | 0.67 (0.53) | 1.00 | 0.07 | 65.62 |
| | Pattern 2 | 3 (5) | 0.15 (0.25) | 0.20 (0.20) | 0.57 (0.44) | 0.67 (0.57) | 1.00 | 0.07 | 65.62 |
| | Pattern 3 | 4 (6) | 0.20 (0.35) | 0.20 (0.20) | 0.50 (0.36) | 0.62 (0.50) | 1.00 | 0.00 | 71.65 |
| | Pattern 4 | 3 (5) | 0.15 (0.30) | 0.20 (0.20) | 0.57 (0.40) | 0.67 (0.53) | 1.00 | 0.07 | 65.62 |
| **2. DWD climate data with 6 continuous variables** | | | | | | | | | |
| PC | wo **PK**(Baseline A) | 9 | 0.20 | 0.83 | 0.14 | 0.15 | 0.90 | 0.22 | 69.32 |
| | Pattern 0 (Baseline B) | 5 (8) | 0.03 (0.20) | 0.67 (0.33) | 0.67 (0.40) | 0.44 (0.50) | 0.71 | 0.36 | 32.70 |
| | Pattern 1 | 5 (9) | 0.03 (0.23) | 0.67 (0.33) | 0.67 (0.36) | 0.44 (0.47) | 0.71 | 0.36 | 32.70 |
| | Pattern 2 | 5 (8) | 0.03 (0.20) | 0.67 (0.33) | 0.67 (0.40) | 0.44 (0.50) | 0.71 | 0.36 | 32.70 |
| Exact Search | wo **PK**(Baseline A) | 6 | 0.20 | 0.17 | 0.45 | 0.59 | 0.91 | 0.28 | 92.87 |
| | Pattern 0 (Baseline B) | 5 (8) | 0.10 (0.20) | 0.33 (0.33) | 0.57 (0.40) | 0.62 (0.50) | 0.98 | 0.12 | 58.38 |
| | Pattern 1 | 5 (5) | 0.10 (0.13) | 0.33 (0.17) | 0.57 (0.56) | 0.62 (0.67) | 0.91 | 0.19 | 57.73 |
| | Pattern 2 | 6 (9) | 0.13 (0.23) | 0.33 (0.33) | 0.50 (0.36) | 0.57 (0.47) | 0.91 | 0.20 | 63.58 |
| DirectLiNGAM | wo **PK**(Baseline A) | 10 | 0.33 | 0.67 | 0.17 | 0.22 | 1.00 | 0.00 | 99.53 |
| | Pattern 0 (Baseline B) | 4 (8) | 0.07 (0.20) | 0.33 (0.33) | 0.67 (0.40) | 0.67 (0.50) | 1.00 | 0.00 | 52.67 |
| | Pattern 1 | 8 (8) | 0.10 (0.10) | 0.83 (0.83) | 0.25 (0.25) | 0.20 (0.20) | 0.64 | 0.43 | 38.03 |
| | Pattern 2 | 4 (7) | 0.03 (0.17) | 0.50 (0.33) | 0.75 (0.44) | 0.60 (0.53) | 0.98 | 0.09 | 40.80 |
| | Pattern 3 | 5 (6) | 0.10 (0.13) | 0.33 (0.33) | 0.57 (0.50) | 0.62 (0.57) | 0.93 | 0.16 | 57.90 |
| | Pattern 4 | 5 (6) | 0.10 (0.17) | 0.33 (0.16) | 0.57 (0.50) | 0.62 (0.62) | 0.92 | 0.18 | 57.80 |
| **3. Sachs' protein data with 11 continuous variables** | | | | | | | | | |
| PC | wo **PK**(Baseline A) | 24 | 0.16 | 0.47 | 0.38 | 0.44 | 0.99 | 0.05 | 294.15 |
| | Pattern 0 (Baseline B) | 15 (19) | 0.04 (0.11) | 0.58 (0.47) | 0.67 (0.48) | 0.52 (0.50) | 0.89 | 0.16 | 166.91 |
| | Pattern 1 | 25 (43) | 0.17 (0.53) | 0.68 (0.16) | 0.26 (0.23) | 0.29 (0.36) | 0.97 | 0.11 | 284.58 |
| | Pattern 2 | 23 (23) | 0.13 (0.24) | 0.74 (0.32) | 0.28 (0.35) | 0.27 (0.46) | 0.97 | 0.09 | 231.27 |
| Exact Search | wo **PK**(Baseline A) | 31 | 0.26 | 0.68 | 0.18 | 0.23 | 0.99 | 0.07 | 374.35 |
| | Pattern 0 (Baseline B) | 17 (19) | 0.07 (0.11) | 0.58 (0.47) | 0.53 (0.48) | 0.47 (0.50) | 0.91 | 0.16 | 202.97 |
| | Pattern 1 | 17 (15) | 0.05 (0.06) | 0.68 (0.58) | 0.55 (0.57) | 0.40 (0.48) | 0.87 | 0.20 | 158.26 |
| | Pattern 2 | 16 (14) | 0.11 (0.14) | 0.53 (0.32) | 0.45 (0.48) | 0.46 (0.57) | 0.95 | 0.12 | 257.53 |
| DirectLiNGAM | wo **PK**(Baseline A) | 29 | 0.25 | 0.47 | 0.28 | 0.36 | 1.00 | 0.01 | 410.23 |
| | Pattern 0 (Baseline B) | 17 (19) | 0.07 (0.11) | 0.53 (0.47) | 0.56 (0.48) | 0.51 (0.50) | 0.91 | 0.16 | 203.02 |
| | Pattern 1 | 15 (13) | 0.07 (0.08) | 0.47 (0.37) | 0.59 (0.60) | 0.56 (0.62) | 0.88 | 0.18 | 220.32 |
| | Pattern 2 | 22 (21) | 0.14 (0.18) | 0.63 (0.47) | 0.33 (0.36) | 0.35 (0.43) | 0.70 | 0.31 | 269.76 |
| | Pattern 3 | 22 (23) | 0.15 (0.18) | 0.53 (0.53) | 0.38 (0.33) | 0.42 (0.39) | 0.92 | 0.16 | 301.51 |
| | Pattern 4 | 20 (16) | 0.12 (0.16) | 0.53 (0.26) | 0.43 (0.47) | 0.45 (0.57) | 0.89 | 0.19 | 264.97 |

4 is inferior to that of Pattern 0. One of the possible reasons for this may be the pruning of the candidate edges suggested from **PK** in the SCD process. For elucidating this behavior, further research is required on what type and amount of information in SCP can truly maximize the performance of SCD.

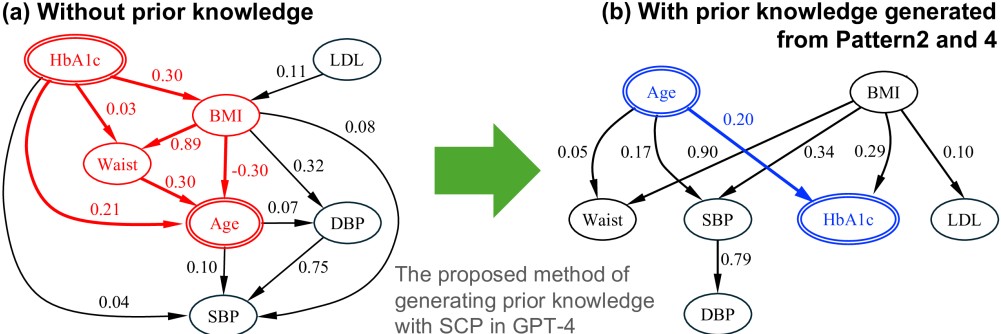

Figure 2: Results of DirectLiNGAM in the health screening data. (a) Result without prior knowledge. (b) Result with prior knowledge, which is generated from GPT-4 with SCP in Patterns 2 and 4. In this randomly selected subset, the DirectLiNGAM result without prior knowledge exhibits unnatural paths drawn in red in (a), which indicates that "Age" is influenced by "HbA1c." However, using the proposed method, the unnatural behavior is clearly mitigated with the guide of prior knowledge generated from GPT-4 with SCP, including the value of causal coefficients in (a) or the bootstrap probabilities of the emergence of directed edges.

## 4.2 Results in Randomly Selected Sub-sample of Health Screening Data Excluded from GPT-4 Pre-Training Dataset

It is difficult to assert that this improvement is solely due to the LLM's high performance in KBCI from the experimental results on open benchmark datasets, because it cannot be determined whether the improvement stems from the LLM's recall of the data obtained during the pre-training process on these datasets. Furthermore, assuming realistic situations, it is also important to confirm the robust effectiveness of this method, even if the range of the available dataset for statistical causal inference is limited to observation data, which may be statistically biased, and the trivial causal relationships are not apparent in SCD without prior knowledge. Therefore, we also apply the proposed methods on the sub-dataset of health screening results, which has been randomly sampled from the entire dataset[12], and the natural ground truth is not presented in the SCD results without **PK**.

In Figure 2(a), the result of DirectLiNGAM without **PK** is shown, and unnatural directed edges to "Age" from other variables are suggested, although the parts of the ground truths from expert knowledge are reversed relationships from these edges, such as "Age"→"HbA1c". However, when the causal discovery is assisted with **PK** generated from GPT-4 with SCP in Patterns 2 and 4, the causal graph becomes more natural: "Age" is not influenced by other variables, and the ground truth "Age"→"HbA1c", which cannot be detected without **PK** in this randomly selected subset, appears in the causal graph, as shown in Figure 2(b). Because this sub-dataset and the analysis results are not disclosed and have been completely excluded from the pre-training data for GPT-4, GPT-4 cannot respond to prompts asking for the causal relationships merely by reproducing the knowledge acquired from the same data. Based on the above, it is verified that the assist of GPT-4 with SCP can cause the result of SCD to further approach the ground truths to an extent, even when the dataset is not learned by GPT-4 and contains bias. Moreover, the effectiveness of the proposed method demonstrated on both open benchmark datasets and unpublished health-screening datasets also supports the validity and applicability of existing works (Ban et al., 2023; Vashishtha et al., 2023; Khatibi et al., 2024), where knowledge-based improvements with LLMs of the causal graph on open benchmark datasets have been demonstrated using their own methods with unique originality. Our research, by showing versatility across both open and unpublished data, further enhances the validity of these existing works.

For further clarity regarding the behaviors of SCP, the mean probabilities of the positive response on the causal relationships from GPT-4 for ground truths in the health screening data, in addition to the statistics of the fitting results with the structure suggested by DirectLiNGAM with SCP are presented in Table 4. In Pattern 0 without SCP, the existence of "Age"→"BMI" and "Age"→"HbA1c" is supported in the probabilities over 0.90, and the probabilities decrease in other patterns with SCP. On the other hand, the opposite behavior

---

[12]The details of the sampling method for this experiment are presented in Appendix C

Table 4: Quantitative evaluation of characteristic results of SCD in the proposed methods on the subset of health screening data with certain biases.
A: The elements of $\boldsymbol{P}$ generated in each prompting pattern for all the appropriate ground truth causal relationships in these variables. The values enclosed in parentheses are the bootstrap probabilities of the directed edges in the DirectLiNGAM algorithm without $\boldsymbol{PK}$. It is suggested that the mean probabilities of the positive response on the causal relationships from GPT-4 are influenced by the results of SCD and bootstrap probabilities when they are included in SCP. Moreover, all the probabilities of reversed, unnatural causal relationships in which "age" is influenced by other factors are extremely closed to zero.
B: CFI, RMSEA and BIC evaluated on the model fitting with the structure discovered by DirectLiNGAM, under the assumption of linear-Gaussian data. The values enclosed in parentheses are the statistics of the structure calculated without $\boldsymbol{PK}$.
It can be inferred that SCP with bootstrap probabilities in Patterns 2 and 4, have enhanced the confidence in "Age"→"DBP" by GPT-4, and improved the BIC when compared with Pattern 0.

| | | | Pattern 0 | Pattern 1 | Pattern 2 | Pattern 3 | Pattern 4 |
|---|---|---|---|---|---|---|---|
| **A. Probability of reproducing ground truth from GPT-4** | **"Age"→"BMI"** (0.166) | | 0.901 | 0.076 | 0.093 | 0.306 | 0.037 |
| | **"Age"→"SBP"** (0.550) | | 0.626 | 0.302 | 0.207 | 0.235 | 0.795 |
| | **"Age"→"DBP"** (0.308) | | 0.001 | 0.019 | 0.115 | 0.095 | 0.926 |
| | **"Age"→"HbA1c"** (0.327) | | 0.986 | 0.170 | 0.723 | 0.046 | 0.176 |
| **B. Statistics of linear-Gaussian fitting with the results of DirectLiNGAM** | **CFI↑** | (1.002) | 0.999 | 0.992 | 0.995 | 0.986 | 0.995 |
| | **RMSEA↓** | (0.000) | 0.018 | 0.054 | 0.032 | 0.057 | 0.032 |
| | **BIC↓** | (124.332) | 103.581 | 89.738 | 89.740 | 103.506 | 89.740 |

is also observed on "Age"→"DBP," which is strongly denied from the GPT-4 domain knowledge without SCP in Pattern 0. It is reasonable to interpret that these probability changes with SCP in GPT-4 are induced by the results of SCD and bootstrap probabilities.

For example, focusing on the relationship "Age"→"BMI," the lack of the direct edge in the SCD result with $\boldsymbol{PK}$ shown in Figure 2(b) and the relatively low bootstrap probability 0.166 in Table 4, may be the cause of lower probability in the SCP patterns than Pattern 0. By contrast, although the hypothesis of "Age"→"DBP" is not supported by the GPT-4 domain knowledge in Pattern 0, the appearance of the edge in Figure 2(a) and the bootstrap probability 0.308 are considered to be the cause of the increase in probability through SCP. Considering the aforementioned properties, the probability that the judgement rendered by GPT-4 regarding causal relationships can be influenced by the SCD results, particularly when the dataset with some significant biases is used in the causal discovery.

Finally, it is also confirmed that the BIC becomes smaller with the assistance by GPT-4 than without $\boldsymbol{PK}$. In particular, the values in Patterns 1, 2, and 4 are lower than Pattern 0. This suggests that SCP can contribute to the discovery of the causal structure with more adequate statistical models.

## 5 Conclusion

In this study, a novel methodology of causal inference, in which SCD and LLM-KBCI are synthesized with SCP and prior knowledge augmentation was developed and demonstrated.

It has been revealed that GPT-4 can cause the output of LLM-KBCI and the SCD result with prior knowledge from LLM-KBCI to approach the ground truth, and that the SCD result can be further improved, if GPT-4 undergoes SCP. Furthermore, with an unpublished real-world dataset, we have demonstrated that GPT-4 with SCP can assist in SCD with respect to domain expertise and statistics, and that the proposed method is effective even if the dataset has never been included in the training data of the LLM and the sample size is not sufficiently large to obtain reasonable SCD results.

**Limitations of This Work**

We have fixed the LLM in our experiment to GPT-4 for its extensive general knowledge and capabilities in common-sense reasoning, and our experiments on several real-world datasets, typically within the realm of common-sense reasoning, have illustrated the potential utility of the proposed method across various scientific domains. We expect that the proposed method will work even if other LLMs are adopted, since the capability of LLM-KBCI has already been demonstrated in several models (Zečević et al., 2023). Considering the continuous emergence of new LLMs and the rapid update of existing LLMs, it is expected that the universal effectiveness of the proposed method across different LLMs will be verified.

Moreover, it is also important to recognize the risks of using current LLMs in our methods: hallucinations at Steps 2 and 3 in Figure 1 can worsen the results of LLM-KBCI and SCD. Considering that hallucinations arise from factors such as memorization of training data (McKenna et al., 2023; Carlini et al., 2023), statistical biases (Jones & Steinhardt, 2022; McMilin, 2022), and predictive uncertainty (Yang et al., 2023; Lin et al., 2024), it is essential to be especially careful when our method is applied in highly specialized academic domains (George & Stuhlmueller, 2023; Pal et al., 2023). To reduce these risks when the proposed method is applied in domains with deeper specificity in the future, it is necessary to further improve our method to more effectively avoid hallucinations. Although we have already introduced ZSCoT (Kojima et al., 2022) and generated knowledge prompting (Liu et al., 2022), employing optimal LLMs that are specifically pre-trained with materials on specific domains is one possible solution, or incorporating techniques such as fine-tuning and RAG could be necessary. Therefore, systematic research in the context of optimal LLM-KBCI is also required on the selection of the optimal LLMs for each domain and on the techniques for utilizing the LLMs.

Furthermore, for generalization of the method we have proposed, we also have to remark that although SCP is indeed likely to enhance the performance of SCD more than the prompting without SCP, whether this improvement is observed can depend on the datasets used for SCD. Moreover, there remains a discussion on the potential for overfitting in SCP, especially with small or biased datasets. Although we have shown the effectiveness of SCP on a sub-sample of the health screening dataset, which contains bias, it is still an open question what conditions of the LLMs or prompting techniques can guarantee robustness for small or biased datasets. In terms of the more reliable application of the proposed method, further basic research on the effect of causal coefficients or bootstrap probabilities on the results of LLM-KBCI is required.

**Broader Impact Statement**

This paper proposed a novel approach that integrates SCD methods with LLMs. This research has the potential to contribute to more accurate causal inference in fields where understanding causality is crucial, such as healthcare, economics, and environmental science. However, the use of LLMs such as GPT-4 necessitates the extensive consideration of data privacy and biases. Moreover, it should also be noted that incorrect causal inferences, caused by hallucinations, biased training data, or inappropriate use of LLMs, could lead to significant consequences for society. In particular, these risks arise when the proposed method is used for completely "black-box" decision making. To prevent these risks, the transparency and interpretability of the whole system should be clarified.

This study highlights the responsible use of artificial intelligence, considering ethical implications and societal impacts. With appropriate guidelines and ethical standards, the proposed methodology can advance scientific understanding and provide extensive widespread benefits to society.

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

# A   Ethics Review

**Ethical Considerations in Methodology and AI Use**   This paper proposes a novel approach that integrates SCD with LLMs. We have thoroughly considered the issues of data privacy and biases associated with the use of LLMs. The proposed methodology enhances the accuracy and efficiency of causal discovery; however, it does not introduce explicit ethical implications beyond those generally applicable to machine learning. We are committed to the responsible use of AI and welcome the scrutiny of the ethics review committee.

**Institutional Review and Consent Compliance of Health Screening Data**   The institutional review board approved this study, and in accordance with the double-blind review process of TMLR, the name of the institution has been withheld. The identity of the institution will be disclosed upon successful completion of the peer review. As we only analyzed anonymized data from the database, the need for informed consent was waived.

# B    Detail of Contents in Each Prompting Pattern

In this section, the details of the prompting in each pattern are presented. For Pattern 0, another prompting template is detailed instead of the sentences shown in Table 1. Moreover, for Patterns 1, 2, 3, and 4, the contents filled in ⟨blank 5⟩ and ⟨blank 9⟩ of the prompt template for SCP shown in Table 1 are described.

**For Pattern 0**    Compared with other patterns of SCP, Pattern 0 does not include any results of SCD without prior knowledge. As a result, the prompt template in Table 1 is not suitable for Pattern 0, as it includes the blanks filled with the description of the dataset and SCD result. Thus, we prepare another prompt template for Pattern 0, which is completely independent of the SCD result, and require GPT-4 to generate the response solely from its domain knowledge. Table 5 presents the prompt template in Pattern 0, which is composed mainly from the ZSCoT concept. Because it does not include information on the SCD method and relies solely on the domain knowledge in GPT-4, the probability matrix from the process of GPT-4 in Pattern 0 is applied independently of the SCD method.

Table 5: The prompt template of $Q_{ij}^{(1)}(x_i, x_j)$ for Pattern 0 for the generation of the expert knowledge of the causal effect from $x_j$ on $x_i$. The "blanks" enclosed with ⟨ ⟩ are filled with description words of the theme of the causal inference and variable names.

| **Prompt Template of $q_{ij}^{(1)} = Q_{ij}^{(1)}(x_i, x_j)$ in Pattern 0 (for all SCD methods)** |
|---|
| We want to carry out causal inference on ⟨blank 1. theme⟩, considering ⟨blank 2. The description of all variables⟩ as variables. |
| If ⟨blank 7. The name of $x_j$⟩ is modified, will it have a direct impact on ⟨blank 8. The name of $x_i$⟩? |
| Please provide an explanation that leverages your expert knowledge on the causal relationship between ⟨blank 7. The name of $x_j$⟩ and ⟨blank 8. The name of $x_i$⟩. 
 Your response should consider the relevant factors and provide a reasoned explanation based on your understanding of the domain. |

**For Patterns 1–4 (in the case of Exact Search and DirectLiNGAM)**    Following the concept of each SCP pattern, the contents filled in ⟨blank 5⟩ shown in Table 1 are summarized in Table 6. In this table, the names of the causes and effected variables are represented as ⟨ cause $i$ ⟩ and ⟨ effect $i$ ⟩ respectively, and the bootstrap probability of this causal relationship in SCD $P_i$ and the causal coefficient of LiNGAM $b_i$ can be included in Patterns 2–4. In Patterns 2 and 4, only the causal relationships with $P_i \neq 0$ are listed in ⟨ blank 5 ⟩. In Pattern 3, the causal relationships with $b_i \neq 0$ are listed in ⟨ blank 5 ⟩.

The contents filled in ⟨blank 6⟩ and ⟨blank 9⟩ also depend on the SCP patterns, and are shown in Table 7. Here, we define the bootstrap probability of $x_j \rightarrow x_i$ as $P_{ij}$. we also define the causal coefficient of $x_j \rightarrow x_i$ in LiNGAM as $b_{ij}$, because the structural equation of LiNGAM is usually defined as[13]:

$$x_i = \sum_{i \neq j} b_{ij} x_j + e_i \tag{1}$$

**Prompt Template in case of PC algorithm**    Although the causal relationships are ultimately represented only as directed edges in Exact Search and DirectLiNGAM, the situation changes slightly when we adopt the PC algorithm along with the codes in "causal-learn." This is because the PC algorithm can also output undirected edges, when the causal direction between a pair of variables cannot be determined. Therefore, we have tentatively decided to include not only directed edges but also undirected edges in SCP. Additionally, we have prepared another prompting template for SCP in the case of PC, as shown in Table 8. This template is

---

[13]Here, the error distribution function $e_i$ is also assumed to be non-Gaussian.

Table 6: Contents filled in ⟨blank 5⟩ shown in Table 1, when Exact Search or DirectLiNGAM is adopted for the SCD process.

| SCP Pattern | Content in ⟨blank 5⟩ |
|---|---|
| **Pattern 1**
Directed edges | All of the edges suggested by the statistical causal discovery are below:
⟨ cause 1 ⟩ → ⟨ effect 1 ⟩
⟨ cause 2 ⟩ → ⟨ effect 2 ⟩
⋮ |
| **Pattern 2**
Bootstrap probabilities
of directed edges | All of the edges with non-zero bootstrap probabilities
suggested by the statistical causal discovery are below:
⟨ cause 1 ⟩ → ⟨ effect 1 ⟩ (bootstrap probability = $P_1$)
⟨ cause 2 ⟩ → ⟨ effect 2 ⟩ (bootstrap probability = $P_2$)
⋮ |
| **Pattern 3**
(DirectLiNGAM Only)
Non-zero causal coefficients
of directed edges | All of the edges and their coefficients of the structural causal model
suggested by the statistical causal discovery are below:
⟨ cause 1 ⟩ → ⟨ effect 1 ⟩ (coefficient = $b_1$)
⟨ cause 2 ⟩ → ⟨ effect 2 ⟩ (coefficient = $b_2$)
⋮ |
| **Pattern 4**
(DirectLiNGAM Only)
Non-zero causal coefficients
and bootstrap probabilities
of directed edges | All of the edges with non-zero bootstrap probabilities
and their coefficients of the structural causal model
suggested by the statistical causal discovery are below:
⟨ cause 1 ⟩ → ⟨ effect 1 ⟩ (coefficient = $b_1$, bootstrap probability = $P_1$)
⟨ cause 2 ⟩ → ⟨ effect 2 ⟩ (coefficient = $b_2$, bootstrap probability = $P_2$)
⋮ |

Table 7: Contents filled in ⟨blank 6⟩ and ⟨blank 9⟩ shown in Table 1, when Exact Search or DirectLiNGAM is adopted for the SCD process.

| SCP Pattern | Case Classification | ⟨blank 6⟩ | Content in ⟨blank 9⟩ |
|---|---|---|---|
| **Pattern 1**
Directed edges | $x_j \to x_i$ emerged | a | -
(No values are filled in) |
| | $x_j \to x_i$ not emerged | no | -
(No values are filled in) |
| **Pattern 2**
Bootstrap probabilities
of directed edges | $P_{ij} \neq 0$ | a | with a bootstrap probability of $P_{ij}$ |
| | $P_{ij} = 0$ | no | -
(No values are filled in) |
| **Pattern 3**
(DirectLiNGAM Only)
Non-zero causal coefficients
of directed edges | $b_{ij} \neq 0$ | a | with a causal coefficient of $b_{ij}$ |
| | $b_{ij} = 0$ | no | -
(No values are filled in) |
| **Pattern 4**
(DirectLiNGAM Only)
Non-zero causal coefficients
and bootstrap probabilities
of directed edges | $P_{ij} \neq 0$ and $b_{ij} \neq 0$ | a | with a bootstrap probability of $P_{ij}$,
and the coefficient is likely to be $b_{ij}$ |
| | $P_{ij} \neq 0$ and $b_{ij} = 0$ | a | with a bootstrap probability of $P_{ij}$,
but the coefficient is likely to be 0 |
| | $P_{ij} = 0$ | no | -
(No values are filled in) |

slightly modified from the one in Table 1. The description for ⟨ blank 5 ⟩ in each SCP pattern is augmented by Table 9, and the description for ⟨ blank 6 ⟩ is similarly augmented by Table 10.

As shown in Table 9, the directed and undirected edges are separately listed and clearly distinguished by the edge symbols of "→" for directed edges and "—" for undirected edges. These are then filled in ⟨ blank 5 ⟩. The pairs of variables connected by undirected edges are represented as ⟨ cause or effect $i$-1 ⟩ and ⟨ cause or effect $i$-2 ⟩, and the bootstrap probability of the emergence of these relationships is represented as $P_i^u$. On the other hand, the bootstrap probability of ⟨ cause $i$ ⟩ → ⟨ effect $i$ ⟩ is represented as $P_i^d$.

The division of the descriptions in ⟨ blank 6 ⟩ is shown in Table 10. The bootstrap probabilities of the appearance of $x_j \to x_i$ and $x_j$—$x_i$ are respectively represented as $P_{ij}^d$ and $P_{ij}^u$.

Table 8: The prompt template of $Q_{ij}^{(1)}(x_i, x_j, \hat{G}_0, \boldsymbol{P^d}, \boldsymbol{P^u})$ in case of the PC algorithm. The "blanks" enclosed with $\langle\ \rangle$ are filled with description words considering the theme of the causal inference, variable names, and the SCD result with the PC algorithm.

---

**Prompt Template of $q_{ij}^{(1)} = Q_{ij}^{(1)}(x_i, x_j, \hat{G}_0, \boldsymbol{P^d}, \boldsymbol{P^u})$ for PC**

---

We want to carry out causal inference on ⟨blank 1. The theme⟩, considering ⟨blank 2. The description of all variables⟩ as variables.

First, we have conducted the statistical causal discovery with the PC (Peter-Clerk) algorithm, using a fully standardized dataset on ⟨blank 4. The description of the dataset⟩.

⟨blank 5. Including here the information of $\hat{G}_0$ (for Pattern 1), or $\boldsymbol{P^d}$ and $\boldsymbol{P^u}$ (for Pattern 2). The detail of the contents depends on prompting patterns, both for directed and undirected edges.⟩

According to the results shown above, it has been determined that
⟨blank 6. The detail of the interpretation on whether there is a causal relationship between $x_j$ and $x_i$ from the result shown in blank 5⟩.

Then, your task is to interpret this result from a domain knowledge perspective and determine whether this statistically suggested hypothesis is plausible in the context of the domain.

Please provide an explanation that leverages your expert knowledge on the causal relationship between ⟨blank 7. The name of $x_j$⟩ and ⟨blank 8. The name of $x_i$⟩, and assess the naturalness of this causal discovery result. Your response should consider the relevant factors and provide a reasoned explanation based on your understanding of the domain.

---

Table 9: Contents filled in ⟨blank 5⟩ shown in Table 8.

| SCP Pattern | Content in ⟨blank 5⟩ |
|---|---|
| **Pattern 1** 
 Directed and undirected edges | All of the edges suggested by the statistical causal discovery are below: 
 ⟨ cause 1 ⟩ → ⟨ effect 1 ⟩ 
 ⟨ cause 2 ⟩ → ⟨ effect 2 ⟩ 
 $\vdots$ 
 In addition to the directed edges above, all of the undirected edges suggested by the statistical causal discovery are below: 
 ⟨ cause or effect 1-1 ⟩ — ⟨ cause or effect 1-2 ⟩ 
 ⟨ cause or effect 2-1 ⟩ — ⟨ cause or effect 2-2 ⟩ 
 $\vdots$ |
| **Pattern 2** 
 Bootstrap probabilities 
 of directed and undirected edges | All of the edges with non-zero bootstrap probabilities suggested by the statistical causal discovery are below: 
 ⟨ cause 1 ⟩ → ⟨ effect 1 ⟩ (bootstrap probability = $P_1^d$) 
 ⟨ cause 2 ⟩ → ⟨ effect 2 ⟩ (bootstrap probability = $P_2^d$) 
 $\vdots$ 
 In addition to the directed edges above, all of the undirected edges suggested by the statistical causal discovery are below: 
 ⟨ cause or effect 1-1 ⟩ — ⟨ cause or effect 1-1 ⟩ (bootstrap probability = $P_1^u$) 
 ⟨ cause or effect 2-1 ⟩ — ⟨ cause or effect 2-2 ⟩ (bootstrap probability = $P_2^u$) 
 $\vdots$ |

Table 10: Contents filled in ⟨blank 6⟩ shown in Table 8.

| SCP Pattern | Case Classification | Content in ⟨blank 6⟩ |
|---|---|---|
| **Pattern 1** 
 Directed and undirected edges | $x_j \rightarrow x_i$ | there may be a direct impact of a change in ⟨blank 7. The name of $x_j$⟩ on ⟨blank 8. The name of $x_i$⟩ |
| | $x_j - x_i$ | there may be a direct causal relationship between ⟨blank 7. The name of $x_j$⟩ and ⟨blank 8. The name of $x_i$⟩, although the direction has not been determined |
| | no edge between $x_i$ and $x_j$ | there may be no direct impact of a change in ⟨blank 7. The name of $x_j$⟩ on ⟨blank 8. The name of $x_i$⟩ |
| **Pattern 2** 
 Bootstrap probabilities of directed and undirected edges | $P_{ij}^d \neq 0$ and $P_{ij}^u \neq 0$ | there may be a direct impact of a change in ⟨blank 7. The name of $x_j$⟩ on ⟨blank 8. Thename of $x_i$⟩ with a bootstrap probability of $P_{ij}^d$. In addition, it has also been shown above that there may be a direct causal relationship between ⟨blank 7. The name of $x_j$⟩ and ⟨blank 8. The name of $x_i$⟩ with a bootstrap probability of $P_{ij}^u$, although the direction has not been determined |
| | $P_{ij}^d \neq 0$ and $P_{ij}^u = 0$ | there may be a direct impact of a change in ⟨blank 7. The name of $x_j$⟩ on ⟨blank 8. The name of $x_i$⟩ with a bootstrap probability of $P_{ij}^d$ |
| | $P_{ij}^d = 0$ and $P_{ij}^u \neq 0$ | there may be a direct causal relationship between ⟨blank 7. The name of $x_j$⟩ and ⟨blank 8. The name of $x_i$⟩ with a bootstrap probability of $P_{ij}^u$, although the direction has not been determined |
| | $P_{ij}^d = 0$ and $P_{ij}^u = 0$ | there may be no direct impact of a change in ⟨blank 7. The name of $x_j$⟩ on ⟨blank 8. The name of $x_i$⟩ |

## C   Details of Datasets used in Demonstrations

In this section, the details of the dataset used in the main body and the appendix are clarified. The ground truths set for the evaluation of the SCD and LLM-KBCI results for each dataset are presented.

### C.1   Auto MPG data

Auto MPG data were originally open in the UCI Machine Learning Repository (Quinlan, 1993), and used as a benchmark dataset for causal inference (Spirtes et al., 2010; Mooij et al., 2016). This dataset consists of the variables around the fuel consumption of cars. We adopt five variables: "Weight", "Displacement", "Horsepower", "Acceleration" and "Mpg"(miles per gallon). Moreover, the number of points of this dataset in the experiment is 392. The ground truth of causal relationships we adopt in this paper is shown in Figure 3; the original has been shown as the example of the kPC algorithm (Spirtes et al., 2010). The differences from the original study (Spirtes et al., 2010) are presented below:

**(1) Loss of "Cylinders"**   Although there is also a discrete variable of "Cylinders" in the original data (Quinlan, 1993), it is omitted in the experiments to focus solely on the continuous variables.

**(2) Directed edge from "Weight" to "Displacement"**   The "Weight" and "Displacement" are connected with an undirected edge, which indicates that the direction cannot be determined in the kPC algorithm, although a causal relationship between these two variables is suggested. However, it is empirically acknowledged that large and heavy vehicles use engines with larger displacement to provide sufficient power to match their size. Thus, we temporally set the direction of the edge between these two variable as "Weight" → "Displacement."

We also recognize that another ground truth was interpreted in the process of reconstructing the Tübingen database for causal-effect pairs (Mooij et al., 2016) [14], and "Mpg" and "Acceleration" were interpreted as effected variables from other elements. This ground truth seems to be reliable, if we do not significantly discriminate between direct and indirect causal effects and targeting the identification of cause and effect from a pair of variables. However, we adopt the ground truth based on the result from the kPC algorithm, because our target is to approach the true causal graph, including multi-step causal relationships.

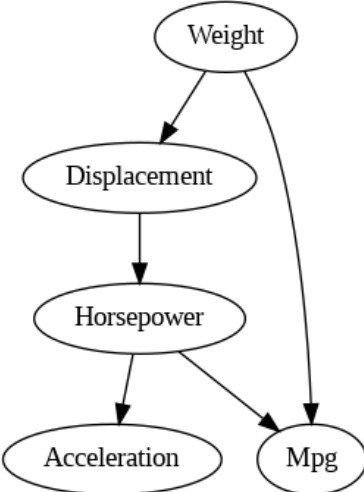

Figure 3: Causal graph of ground truth adopted for Auto MPG data in this study.

In Figure 4, the results of basic causal structure analysis by the PC, Exact Search, and DirectLiNGAM algorithms without prior knowledge are presented. Several reversed edges from ground truths such as "Mpg" → "Weight" are observed.

---

[14]https://webdav.tuebingen.mpg.de/cause-effect/

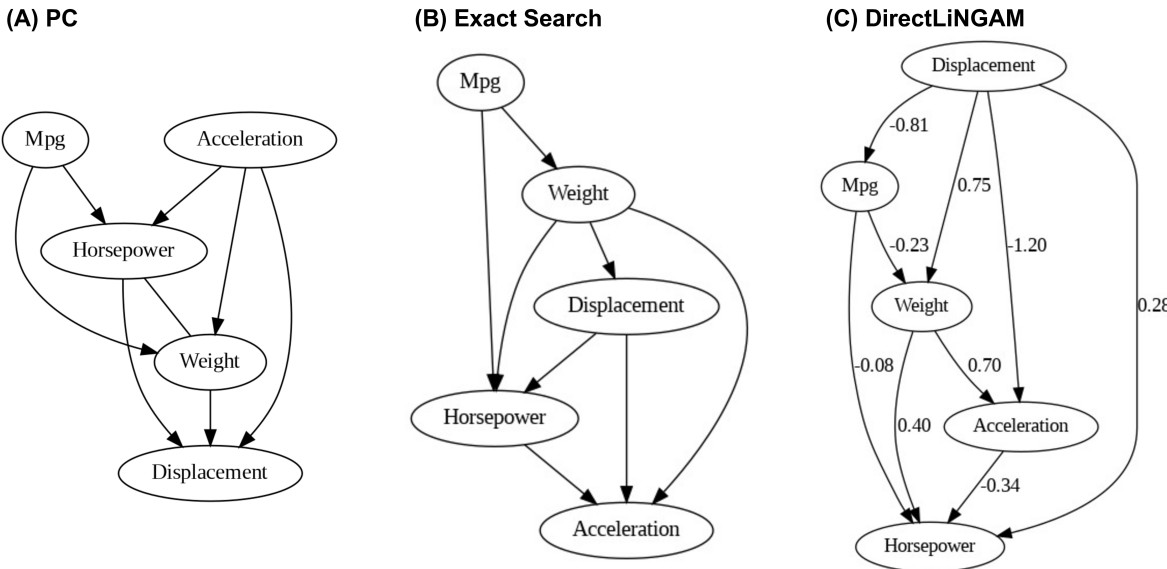

Figure 4: Results of SCD on Auto MPG data with (A) PC, (B) Exact Search, and (C) DirectLiNGAM.

## C.2 DWD climate data

The DWD climate data were originally provided by the DWD [15], and several of the original datasets were merged and reconstructed as a component of the übingen database for causal-effect pairs (Mooij et al., 2016). This dataset consists of six variables: "Altitude", "Latitude", "Longitude", "Sunshine" (duration), "Temperature" and "Precipitation". The number of points of this dataset is 349, which corresponds to the number of weather stations in Germany without missing data.

Because there is no ground truth on this dataset advocated, except for that in the übingen database for causal-effect pairs (Mooij et al., 2016), we adopt it temporally in this experiment, as shown in Figure 5.

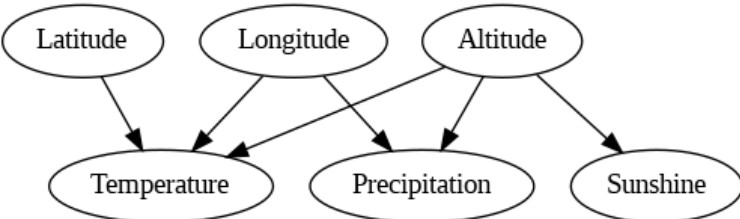

Figure 5: Causal graph of ground truth adopted for DWD climate data in this study.

In Figure 6, the results of basic causal structure analysis by the PC, Exact Search, and DirectLiNGAM algorithms without prior knowledge are presented. In all the causal graphs in Figure 6, several unnatural behaviors are observed, such as "Altitude" being effected by other climate variables, which we interpret as reversed causal relationships from the ground truths.

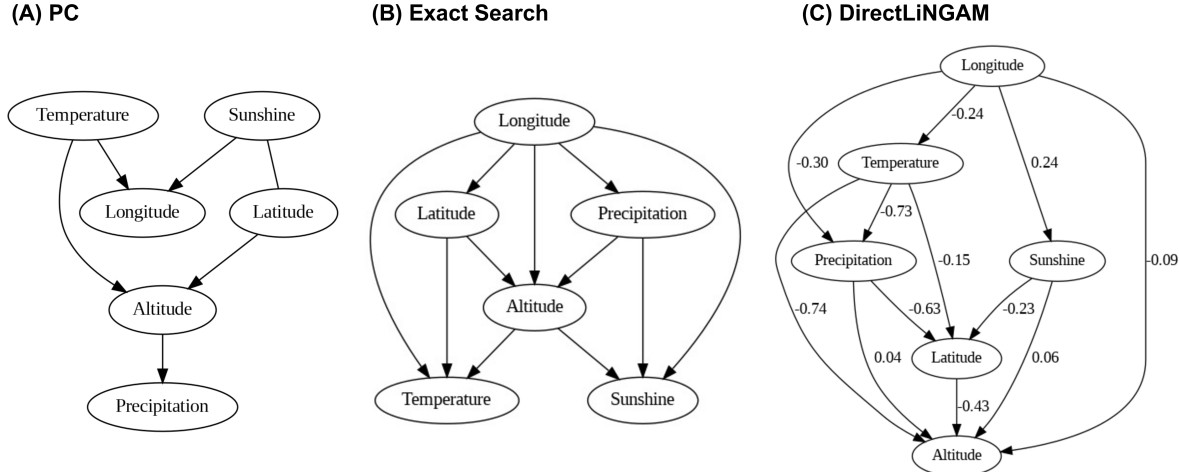

Figure 6: Results of SCD on DWD climate data with (A) PC, (B) Exact Search, and (C) DirectLiNGAM.

---

[15]https://www.dwd.de/

### C.3 Sachs protein data

This dataset consists of the variables of the phosphorylation level of proteins and lipids in primary human immune cells, which were originally constructed and analyzed with the non-parametric causal structure learning algorithm by Sachs et al. (2005). It contains 11 continuous variables: "raf", "mek", "plc", "pip2", "pip3", "erk", "akt", "pka", "pkc", "p38" and "jnk". The number of points of this dataset is 7466.

The ground truth adopted in this study is almost the same as the interpretation shown in the study by Sachs et al. (2005). The differences from the causal graph visually displayed in the original paper are presented below:

**(1) Reversed edge between "pip3" and "plc"**   Although the directed edge "plc" → "pip3" was detected in the original study, it was denoted as "reversed," which may be the reversed direction from the expected edge. Thus, we adopt the causal relationship of "pip3" → "plc" which Sachs *et al.* anticipated as true from an expert point of view.

**(2) Three missed edges in the original study**   In the study by Sachs *et al.*, "pip2" → "pkc',' "plc" → "pkc," and "pip3" → "akt" did not appear in the Bayesian network inference result, although they were expected to be direct causal relationships from the domain knowledge. We adopt these three edges for the ground truth considering that they may not appear under certain SCD conditions and assumptions.

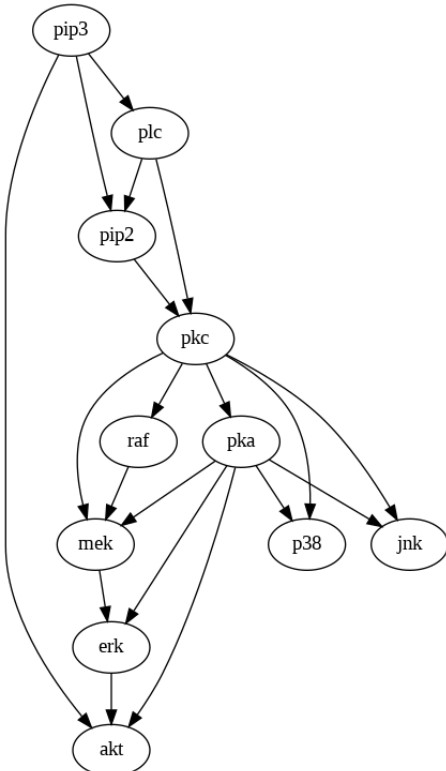

Figure 7: Causal graph of ground truth adopted for Sachs protein data in this study.

In Figure 8, the results of the basic causal structure analysis by the PC, Exact Search, and DirectLiNGAM algorithms without prior knowledge are shown.

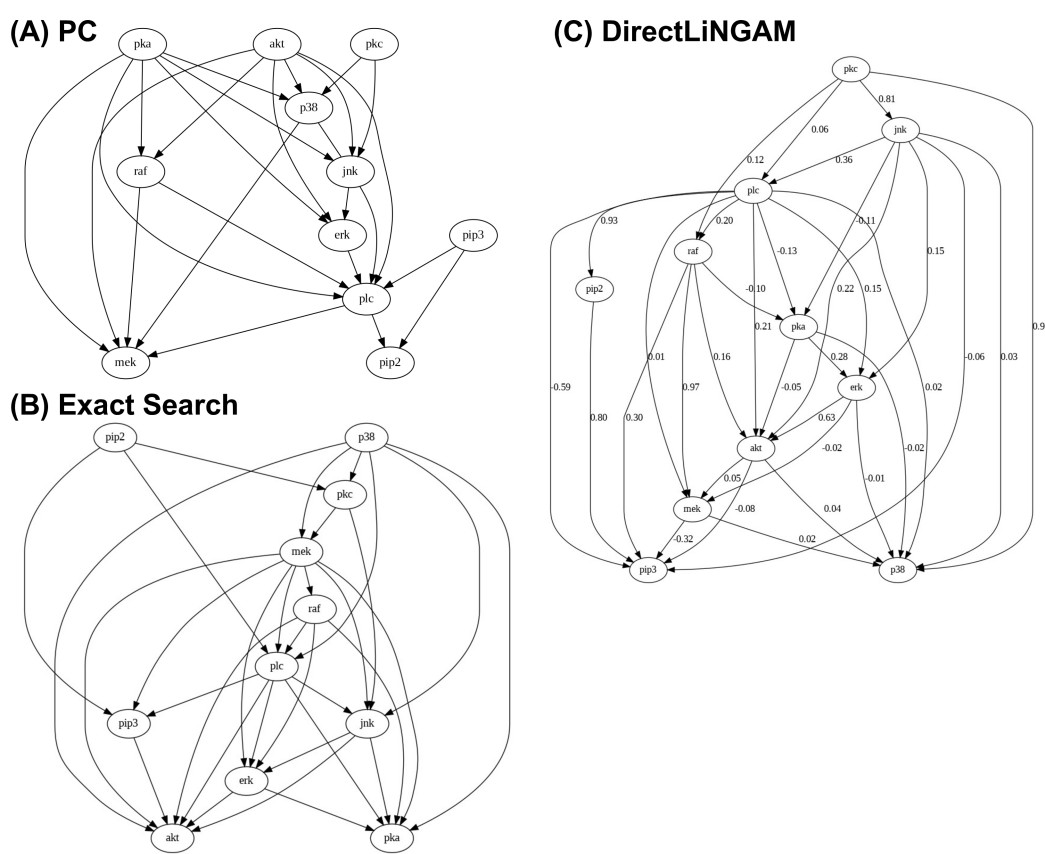

Figure 8: Results of SCD on Sachs data with (A) PC, (B) Exact Search, and (C) DirectLiNGAM.

### C.4  Health screening data (closed data and not included in GPT-4's pre-training materials)

To confirm that GPT-4 can adequately judge the existence of causal relationships with SCP, even if the dataset used in SCD is not included in the pre-training dataset of GPT-4, we have additionally prepared the health screening dataset of workers in engineering and construction contractors, which is not disclosed because of its sensitive nature from personal handling and other private aspects. It contains seven continuous variables: body mass index ("BMI"), waist circumference ("Waist"), Systolic blood pressure ("SBP"), Diastolic blood pressure ("DBP"), hemoglobin A1c ("HbA1c"), low density lipoprotein cholesterol ("LDL"), and age ("Age"). The number of total points of this dataset is 123 151.

Although the causal relationships between all pairs of variables are not completely determined, we set two types of ground truths.

**(1) Directed edges interpreted as ground truths**    We empirically set four directed edges below as ground truths.

- "Age"→"BMI"(Clarke et al., 2008; Alley et al., 2008; Gordon-Larsen et al., 2010; Yang et al., 2021)

- at least one of "Age"→"SBP" and "Age"→"DBP"(Gurven et al., 2012)

- "Age"→"HbA1c"(Pani et al., 2008; RaviKumar et al., 2011; Dubowitz et al., 2014)

**(2) Variable interpreted as a parent for all other variables**    "Age" is an unmodifiable background factor. Furthermore, it has been clearly demonstrated in numerous medical studies that aging affects "BMI," "SBP," "DBP," and "HBA1c." Based on this specialized knowledge, we interpret "Age" as a parent for all other variables.

The ground truths introduced above also appear in the result of DirectLiNGAM without prior knowledge, as shown in Figure 9. Although "Age"→"HbA1c" is confirmed in this result, the causal coefficient of this edge is relatively small. Thus, depending on the number of data points or the bias of the dataset, it is possible that this edge does not appear in all SCD methods without prior knowledge. For the experiment, to confirm that GPT-4 can supply SCD with adequate prior knowledge, even if a direct edge of the ground truth is not apparent, we have repeated the sampling of 1000 points from the entire dataset, until we obtained a subset on which PC, Exact Search, and DirectLiNGAM cannot discern the causal relationship "Age"→"HbA1c" without prior knowledge.

The results of the SCD on the subset are shown in Figure 10, and this subset is adopted to confirm the effectiveness of the proposed method. It is confirmed in all SCD results that "Age" → "HbA1c" does not appear, and "Age" is directly influenced by other variables, which we interpret as an unnatural behavior from the domain knowledge.

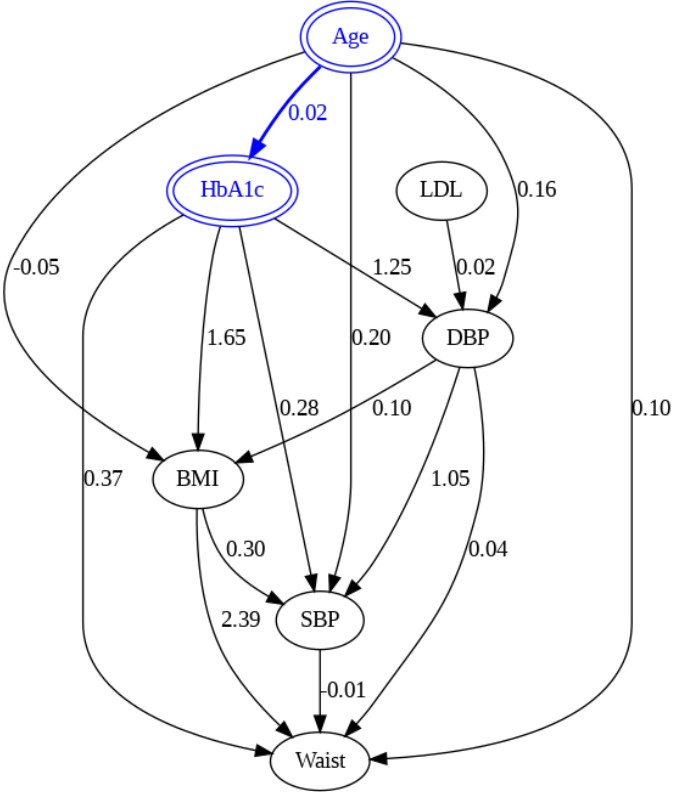

Figure 9: Causal graph suggested by DirectLiNGAM using full points of the health screening data.

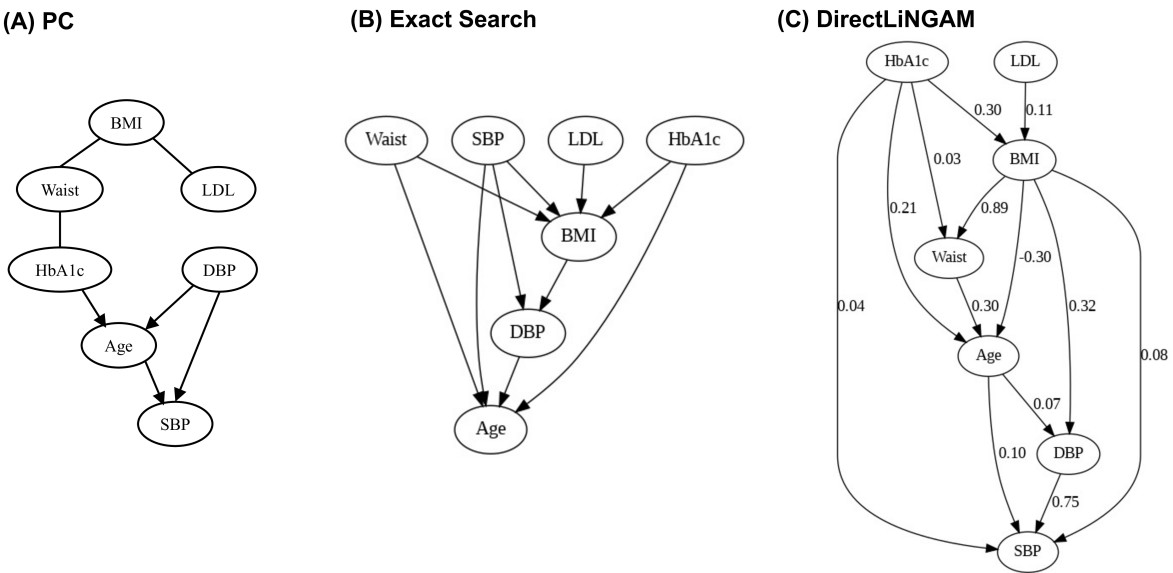

Figure 10: Results of SCD on the randomly selected subsample in the health screening dataset with (A) PC, (B) Exact Search, and (C) DirectLiNGAM.

# D    Token Generation Probabilities as Confidence of Large Language Models and Sensitivity Analysis

In this section, we provide a supplemental explanation of the properties of the mean probability $\overline{p_{ij}}$, that the LLM generates the token "yes" as the response to the prompt asking for the confidence in the causal relationship from $x_j$ to $x_i$, highlighting the properties of the token generation process in Transformer-based LLMs. We also discuss the mean probability $\overline{r_{ij}}$ that the LLM generates the token "no," and explain the symmetric behavior of $\overline{p_{ij}}$ and $\overline{r_{ij}}$ arising from the empirical relationship $\overline{p_{ij}} + \overline{r_{ij}} = 1$, under the assumption of the "faithfulness" of the LLM to the prompt. Moreover, we quantitatively evaluate the fluctuation of $\overline{p_{ij}}$ with the standard error $SE(p_{ij})$ as indices of the sensitivity or reliability of these probabilities measured in our experiments, through the regression analysis with a phenomenological distribution function $SE(\overline{p_{ij}}) = a_p - b_p(\overline{p_{ij}} - 0.5)^2$. Finally, the precision of the classification of forbidden causal paths is evaluated through numerical simulations.

The discussion in this section not only contributes to the evaluation of the LLM's confidence in the causal relationship between pairs of variables, but also supports the reliability of the LLM's outputs.

## D.1    Details of Token Generation Probabilities Calculations

As explained in Section 3.1, we quantitatively evaluate the generation probability of the token "yes" from GPT-4, as the confidence of GPT-4 regarding the existence of the causal relationship. Although the details of GPT-4's architecture have not been disclosed, it is naturally assumed that GPT-4 employs a temperature-adjusted softmax function for token generation, considering the following facts:

- The previous GPT series developed by OpenAI are based on the Transformer architecture (Radford et al., 2019; Brown et al., 2020)

- The temperature hyperparameter can be optionally set

- Log-probabilities of the most likely tokens at each token position can be obtained

In the process of text generation in a Transformer, the model first calculates the logit $z_k$ for token $c_k$ ($0 \leq k \leq K$, where $K$ is the number of candidate tokens) as a likelihood score for all candidate tokens based on the prompted text $X$. Then, the conditional probability distribution of the emergence of token $c_a$ with the logit $z_a$, $P(c_a|X)$, is calculated through the temperature-adjusted softmax function as follows (Tunstall et al., 2022):

$$P(c_a|X, T) = \frac{\exp(z_a/T)}{\sum_{k=0}^{K} \exp(z_k/T)} \tag{2}$$

Here, $T$ is a temperature hyperparameter that adjusts the conditional probability distribution. Finally, the next token is sampled based on the conditional probability distribution above.

In this work, $T$ was fixed to 0.7. In the third step, the log-probability $L_{ij}^{(m)} = L^{(m)}(\epsilon_{\text{GPT4}}(q_{ij}^{(2)}) = \text{"yes"})$ shown in Algorithm 1 is directly sampled for $M$ times using the optional function of GPT-4. Considering that the token generation probability is represented by Eq. (2), we have calculated the mean probability $\overline{p_{ij}}$ as follows:

$$\overline{p_{ij}} = \frac{\sum_{m=1}^{M} \exp(L_{ij}^{(m)})}{M} = \frac{\sum_{m=1}^{M} P^{(m)}\left(\text{"yes"}|q_{ij}^{(2)}, T\right)}{M} \tag{3}$$

We have interpreted $\overline{p_{ij}}$ calculated above as the confidence[16] of GPT-4 in the existence of the causal effect from $x_j$ on $x_i$.

---

[16]Although it is suggested that this interpretation is valid in common-sense reasoning from our experiments, it should be noted that this probability simply represents the likelihood that the LLM selected the expected token as the answer to the prompt. In this sense, this "confidence" can be biased under some conditions since it depends on the architecture and the pre-training datasets of the LLM. Therefore, as discussed in Section 5, it is required to further improve our method in order to avoid hallucinations more reliably, especially for applications in highly specialized academic domains.

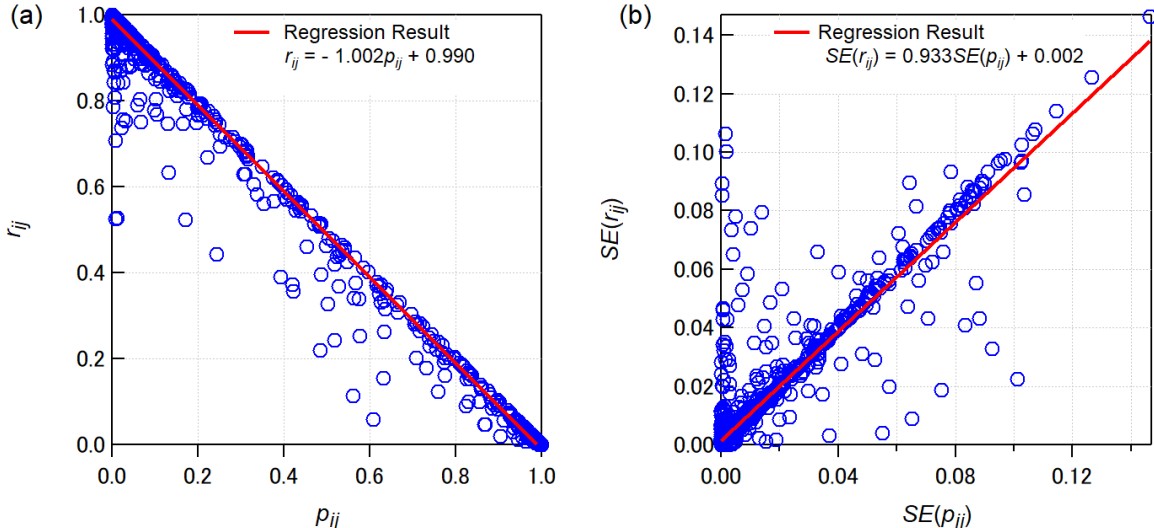

Figure 11: (a) The relationship between $\overline{p_{ij}}$ and $\overline{r_{ij}}$. The red line is the result of least squares regression with $\overline{r_{ij}} = \alpha \overline{p_{ij}} + \beta$. The estimated values ($\pm$ the standard errors) of the coefficients are $\alpha = -1.002 \pm 0.002 (\simeq -1)$ and $\beta = 0.990 \pm 0.001 (\simeq 1)$. (b) The relationship between $SE(\overline{p_{ij}})$ and $SE(\overline{r_{ij}})$. The red line is the result of least squares regression with $SE(\overline{r_{ij}}) = \gamma SE(\overline{p_{ij}}) + \delta$. The estimated values ($\pm$ the standard errors) of the coefficients are $\gamma = 0.933 \pm 0.011 (\simeq 1)$ and $\delta = 0.0015 \pm 0.0002 (\simeq 0)$.

### D.2 Confidence in the Absence of the Causal Relationship

Although the generation probability of the token "no" is not evaluated in the main text, it is valuable to discuss the ideal behavior of this probability for the subsequent discussion of the stability of $\overline{p_{ij}}$. In the same context as $\overline{p_{ij}}$, the mean probability of generating the token "no", $\overline{r_{ij}}$, is calculated as follows:

$$\overline{r_{ij}} = \frac{\sum_{m=1}^{M} P^{(m)}\left(\text{"no"}|q_{ij}^{(2)}, T\right)}{M} \tag{4}$$

Since GPT-4 is required to answer only with "yes" or "no," under the assumption that the response from GPT-4 is faithful to the prompting explained in Table 2, it is expected that $P^{(m)}(\text{"yes"}|q_{ij}^{(2)}, T) + P^{(m)}(\text{"no"}|q_{ij}^{(2)}, T) = 1$. In this section, we define this property as "faithfulness." Therefore, from Eqs. (3) and (4), the ideal relation between $\overline{p_{ij}}$ and $\overline{r_{ij}}$ is expressed as follows:

$$\overline{r_{ij}} = -\overline{p_{ij}} + 1 \tag{5}$$

Since Eq. (5) is also expressed in a symmetrical form such as $\overline{r_{ij}} - 0.5 = -(\overline{p_{ij}} - 0.5)$, this property is expected to contribute to a clearer quantitative interpretation of the stability of $\overline{p_{ij}}$. This is particularly important for validating the latter discussion on the fluctuation of $\overline{p_{ij}}$.

### D.3 Stability of Token Generation Probabilities

For the quantitative evaluation of the stability of $\overline{p_{ij}}$, we compare $\overline{p_{ij}}$ and $\overline{r_{ij}}$, along with their standard errors ($SE(p_{ij})$ and $SE(r_{ij})$) estimated through trials conducted $M = 5$ times. We temporarily concatenate the results of LLM-KBCI on the datasets of AutoMPG, DWD, and Sachs (for all the prompting patterns and for all the SCD algorithms), and on the health-screening data (for all the prompting patterns with

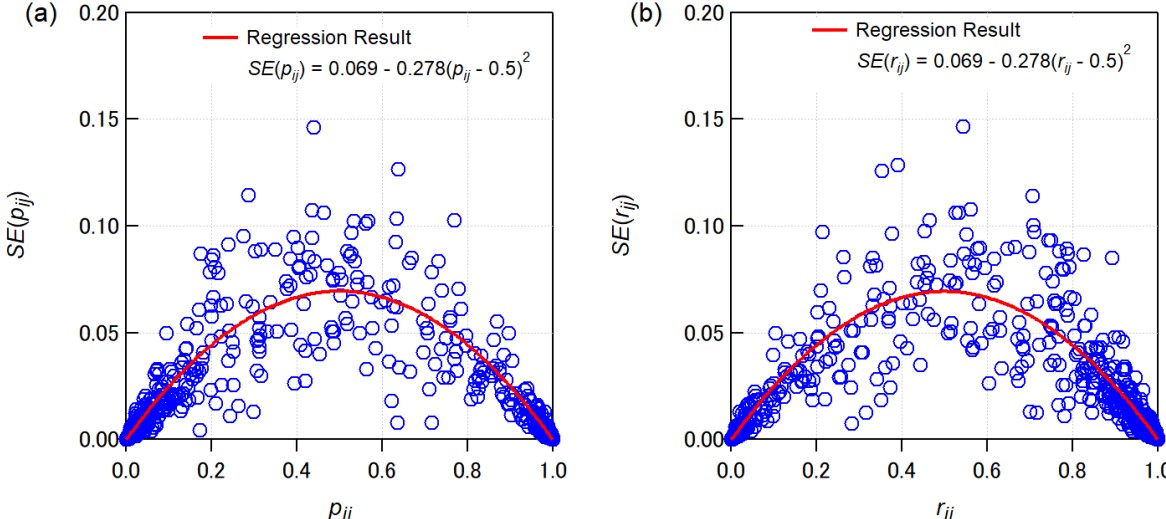

Figure 12: (a) The relationship between the mean probability $\overline{p_{ij}}$ and the standard error $SE(p_{ij})$. The red line is the result of least squares regression with $SE(\overline{p_{ij}}) = a_p - b_p(\overline{p_{ij}} - 0.5)^2$. The estimated values ($\pm$ the standard errors) of the coefficients are $a_p = 0.0694 \pm 0.0007$ and $b_p = 0.2783 \pm 0.0030$. (b) The relationship between the mean probability $\overline{r_{ij}}$ and the standard error $SE(r_{ij})$. The red line is the result of least squares regression with $SE(\overline{r_{ij}}) = a_r - b_r(\overline{r_{ij}} - 0.5)^2$. The estimated values ($\pm$ the standard errors) of the coefficients are $a_r = 0.0692 \pm 0.0007$ and $b_r = 0.2780 \pm 0.0032$.

DirectLiNGAM), under the assumption that the relations among $\overline{p_{ij}}$, $\overline{r_{ij}}$, $SE(p_{ij})$ and $SE(q_{ij})$ are almost independent of the domains. This holds even if SCP is conducted within the realm of common-sense reasoning. The number of points of the concatenated data is 1650.

Figure 11 (a) shows the relation between $\overline{p_{ij}}$ and $\overline{r_{ij}}$, and the red line indicates the result of least squares regression with $\overline{r_{ij}} = \alpha\overline{p_{ij}} + \beta$. Considering that the estimated values of the causal coefficients ($\alpha = -1.002$, $\beta = 0.990$) exhibit similar behavior to Eq. (5), it can be inferred that the assumption of the "faithfulness" of the LLM generation to the prompts is valid in our experiments. Therefore, it seems approximately valid to assume the symmetrical relation $\overline{r_{ij}} - 0.5 = -(\overline{p_{ij}} - 0.5)$.

Moreover, Figure 11 (b) shows the relation between $SE(\overline{p_{ij}})$ and $SE(\overline{r_{ij}})$, where the red line indicates the result of least squares regression with $SE(\overline{r_{ij}}) = \gamma SE(\overline{p_{ij}}) + \delta$. Given the estimated values of the causal coefficients ($\gamma = 0.933$, $\delta = 0.002$), it can be inferred that the approximation $SE(\overline{p_{ij}}) \simeq SE(\overline{r_{ij}})$ holds true. This fact provides further evidence for the approximately symmetric behavior expressed as $\overline{r_{ij}} - 0.5 = -(\overline{p_{ij}} - 0.5)$.

Figure 12 shows the relation between $\overline{p_{ij}}$ and $SE(p_{ij})$ in (a) and the relation between $\overline{r_{ij}}$ and $SE(r_{ij})$ in (b). These two graphs exhibit mutually related characteristics as follows:

- In the limit of $\overline{p_{ij}} \to 1$ or $\overline{r_{ij}} \to 0$, both $SE(p_{ij})$ and $SE(r_{ij})$ approach 0, indicating that $\overline{p_{ij}}$ and $\overline{r_{ij}}$ become highly stable when confidence in the presence of the causal effect from $x_j$ to $x_i$ is very high.

- In the limit of $\overline{p_{ij}} \to 0$ or $\overline{r_{ij}} \to 1$, both $SE(p_{ij})$ and $SE(r_{ij})$ approach 0, indicating that $\overline{p_{ij}}$ and $\overline{r_{ij}}$ become highly stable when confidence in the absence of the causal effect from $x_j$ to $x_i$ is very high.

Furthermore, the entire form of the $\overline{p_{ij}}$ - $SE(p_{ij})$ distribution is quantitatively similar to the $\overline{r_{ij}}$ - $SE(r_{ij})$ distribution. From this fact, it is expected that the functional form of $SE(p_{ij})$ explained with $p_{ij}$ is approximately the same as $SE(r_{ij})$ explained with $r_{ij}$. Therefore, in order to interpret the stability of the probability easily and clearly, we assume a common distribution function between the $\overline{p_{ij}}$ - $SE(p_{ij})$ and $\overline{r_{ij}}$ - $SE(r_{ij})$ distributions, as well as the symmetric behavior shown in Eq. (5), and regress the $\overline{p_{ij}}$ - $SE(p_{ij})$ distribution with a phenomenological function as follows:

$$SE(p_{ij}) = a_p - b_p(\overline{p_{ij}} - 0.5)^2 \tag{6}$$

In the same context, the $\overline{r_{ij}}$ - $SE(r_{ij})$ distribution is regressed with a phenomenological function as follows:

$$SE(r_{ij}) = a_r - b_r(\overline{r_{ij}} - 0.5)^2 \tag{7}$$

Ideally, if the symmetrical behavior holds true, the relationship $(a_p, b_p) = (a_r, b_r)$ is expected. From the analysis results of the least squares regression, the coefficients of Eqs. (6) and (7) are estimated as $(a_p, b_p) = (0.0694, 0.2783)$ and $(a_r, b_r) = (0.0692, 0.2780)$. Thus, it is confirmed that $(a_p, b_p) \simeq (a_r, b_r)$ holds true.

From the results above, the standard error of $p_{ij}$ becomes the maximum value (expected to be 0.062) when $\overline{p_{ij}} = 0.5$. In our experiments, the fluctuation becomes critical at the criteria $\alpha_1$ and $\alpha_2$ explained in Section 3.1. From the phenomenological analysis shown above, the expected standard errors of $\overline{p_{ij}}$ at the criteria ($\alpha_1 = 0.05$ and $\alpha_2 = 0.95$ in our experiments) are estimated through Eq. (6) to be around 0.013. Therefore, when $\overline{p_{ij}}$ is around the threshold of 0.05 or 0.95, the probability fluctuation around this standard error can lead to a different **PK** matrix.

### D.4 Effect on the Decision of Prior Knowledge Matrix

For a more detailed analysis of the effect of the fluctuation on the determination of **PK**, we focus on two indices of sensitivity: (A) the statistically estimated probability that the true value of $p_{ij}$ ($p_{ij}^{true}$) satisfies the inequality $p_{ij}^{true} < \alpha_1$, and (B) the area under the curve (AUC) calculated through a Monte Carlo simulation.

### (A) Statistically Estimated Probability of True Confidence

For a simple simulation, we assume that $SE(p_{ij})$ is calculated with Eq. (6) and $(a_p, b_p) = (0.0694, 0.2783)$ as estimated through the regression analysis in Appendix D.3. Moreover, we assume that the distribution of $p_{ij}^{true}$ is expressed with a Gaussian distribution, with mean value $\overline{p_{ij}}$ and standard deviation $SE(\overline{p_{ij}})$, as described in Eq. (8).

$$f_{p_{ij}^{true}}(\overline{p_{ij}}, SE(\overline{p_{ij}}), p) = \frac{1}{\sqrt{2\pi}SE(\overline{p_{ij}})} \exp\left( -\frac{(p - \overline{p_{ij}})^2}{2SE(\overline{p_{ij}})^2} \right) \tag{8}$$

Under the assumptions above, $P(p_{ij}^{true} < \alpha_1)$, the probability that the true confidence probability $p_{ij}^{true}$ is below the forbidden edge threshold $\alpha_1$, is calculated as follows:

$$P(p_{ij}^{true} < \alpha_1) = \int_0^{\alpha_1} dp f_{p_{ij}^{true}}(\overline{p_{ij}}, SE(\overline{p_{ij}}), p) \tag{9}$$

Then, $P(p_{ij}^{true} < \alpha_1)$ is numerically calculated using Eq. (9), and the result is shown in Fig. 13 (a). It is observed that when $\overline{p_{ij}} \leq 0.04$, $p_{ij}^{true}$ is almost certainly below the threshold $\alpha_1 = 0.05$, and when $\overline{p_{ij}} \geq 0.08$, $p_{ij}^{true}$ is almost certainly above 0.05. However, between $0.04 \leq \overline{p_{ij}} \leq 0.08$ (around 0.05), $P(p_{ij}^{true} < \alpha_1)$ decreases with the increase in $\overline{p_{ij}}$. Although it is confirmed that the width of the region of this decreasing $P(p_{ij}^{true} < \alpha_1)$ is similar in the order of $SE(\overline{p_{ij}}) \simeq 0.013$ around $\overline{p_{ij}} = 0.05$, the slope of the decrease is gentler in the region of $\overline{p_{ij}} \geq 0.05$ than in the region of $\overline{p_{ij}} \leq 0.05$. This asymmetric property originates from the phenomenological nonlinear function of $SE(\overline{p_{ij}})$ as described in Eq. (6). If the threshold of the forbidden edge $\alpha_1$ is decreased, it is possible to shrink the region of decreasing $P(p_{ij}^{true} < \alpha_1)$ and the asymmetric slope. However, although we have tentatively set $\alpha_1$ to 0.05, the optimal threshold is expected to be discussed in

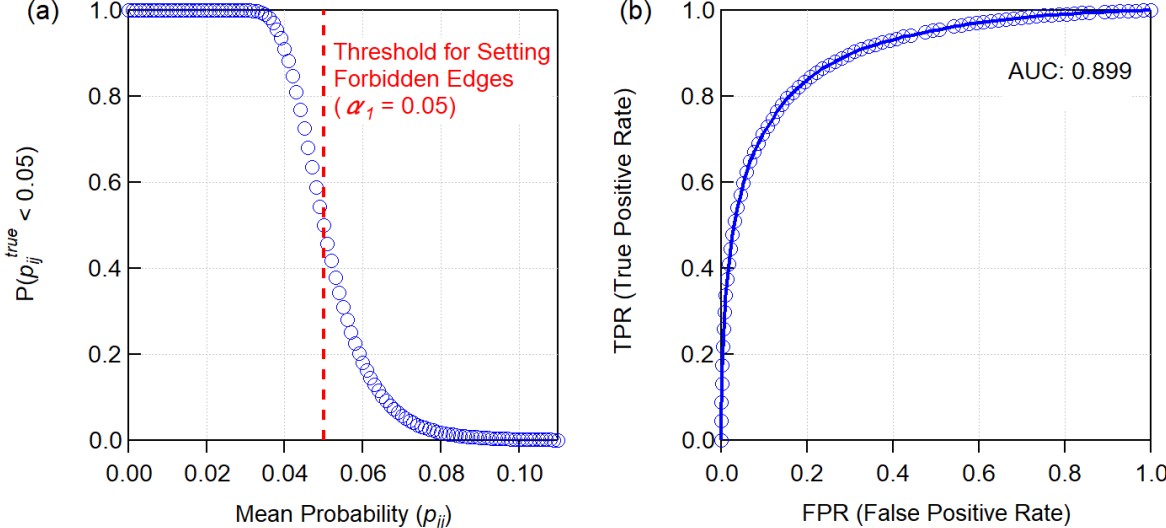

Figure 13: (a) The relationship between calculated $P(p_{ij}^{true} < \alpha_1)$ and $\overline{p_{ij}}$ based on Eq. (9). (b) ROC curve estimated through a Monte Carlo simulation based on Eq. (8). The AUC calculated with this ROC curve is 0.899.

future research from a practical point of view. It should also be noted that a lower $\alpha_1$ leads to a **PK** matrix, which includes more uncertain elements from domain expert knowledge and can be less effective in the fourth step in Fig. 1 than expected.

### (B) ROC and AUC Estimated through a Monte Carlo Simulation

Since $p_{ij}^{true}$ cannot be known in our experiments, it is impossible to construct the receiver operating characteristic (ROC) curve directly with the probability data obtained in our experiments. Instead, we have conducted a Monte Carlo simulation, through which $p_{ij}^{true}$ is randomly generated using the distribution described in Eq. (8). Furthermore, for the calculation of ROC and AUC, we define the conditions for true positive (TP), false positive (FP), true negative (TN), and false negative (FN), in the context of judging whether the causal path from $x_j$ to $x_i$ is forbidden, as follows:

- TP: generated $p_{ij}^{true}$ satisfies $p_{ij}^{true} < \alpha_1$, and experimentally fixed $\overline{p_{ij}}$ satisfies $\overline{p_{ij}} < \alpha_1$.

- FP: generated $p_{ij}^{true}$ satisfies $p_{ij}^{true} \geq \alpha_1$, and experimentally fixed $\overline{p_{ij}}$ satisfies $\overline{p_{ij}} < \alpha_1$.

- TN: generated $p_{ij}^{true}$ satisfies $p_{ij}^{true} \geq \alpha_1$, and experimentally fixed $\overline{p_{ij}}$ satisfies $\overline{p_{ij}} \geq \alpha_1$.

- FN: generated $p_{ij}^{true}$ satisfies $p_{ij}^{true} < \alpha_1$, and experimentally fixed $\overline{p_{ij}}$ satisfies $\overline{p_{ij}} \geq \alpha_1$.

The practical Monte Carlo simulation has been conducted 1000 times for each $\overline{p_{ij}}$ value, varying $\overline{p_{ij}}$ from 0.032 to 0.108[17] in increments of 0.001, and the calculated ROC through this simulation is shown in Fig. 13. AUC is estimated to be 0.899, indicating excellent discrimination of the forbidden direct causal path (Hosmer & Lemeshow, 2000). From this analysis, it is inferred that in the realm of common-sense reasoning, the decision

---

[17]In this region, $P(p_{ij}^{true} < \alpha_1)$ satisfies $0.001 < P(p_{ij}^{true} < \alpha_1) < 0.999$. Therefore, for the practical calculation of ROC and AUC, we have assumed that when $\overline{p_{ij}}$ satisfies $\overline{p_{ij}} < 0.032$, $p_{ij}^{true}$ is under $\alpha_1$ and always classified as TP, and that when $\overline{p_{ij}}$ satisfies $\overline{p_{ij}} > 0.108$, $p_{ij}^{true}$ is $\geq \alpha_1$ and always classified as TN.

process of **PK** from the confidence of GPT-4 at the third step in Fig. 1 according to the interpretation obtained in the second step, is statistically valid.

### D.5 Limitations in Sensitivity Analysis of Token Generation Probabilities

Although we have discussed the sensitivity of the token generation probabilities as confidence of GPT-4 above, there are several limitations in elucidating the entire properties of the fluctuation induced by the randomness of the LLMs. For further research to sophisticate our methods, we introduce some additional points of issue related to the measurement stability in the third step in Fig. 1.

**Sensitivity in the Explanation Obtained in Step 2**

Although we have evaluated the probability fluctuation emergent in Step 3 of Fig. 1, $\overline{p_{ij}}$ and $SE(\overline{p_{ij}})$ can also depend on $a_{ij}$, the output in Step 2, based on the relationship as $q_{ij}^{(2)} = Q_{ij}^{(2)}(q_{ij}^{(1)}, a_{ij})$. Since it is practically impossible to evaluate the fluctuation of $a_{ij}$ at Step 2 quantitatively, we have discussed the token generation probability and its sensitivity with a fixed $a_{ij}$.

Furthermore, we have not discussed in this work the validity of $a_{ij}$, as this is expected to be debated within the domain of research specific to LLMs, which is outside the scope of our study. However, it should be noted that although our method has proven to be effective in many cases through this research, it may be negatively impacted if an LLM makes biased judgments. Therefore, ongoing efforts to eliminate biases in LLMs and prevent hallucinations are required. Additionally, considering the future applications of this method, careful consideration and judgment will continue to be necessary.

**Imperfect Satisfaction of the Condition of the "Faithfulness"**

In our work, the "faithfulness" is almost confirmed as seen in Fig. (11). However, there are several points that are obviously deviated from the regression line, or Eq. (5). Specifically, out of a total of 1650 points in the case of our analysis in this section, approximately 1400 points satisfy the relationship $0.99 < \overline{p_{ij}} + \overline{r_{ij}} < 1$. In contrast, for about 80 points, $\overline{p_{ij}} + \overline{r_{ij}} < 0.95$, indicating clear violations of "faithfulness."

The break of this "faithfulness" means there is a generation probability of the token that is neither "yes" nor "no," at a level that cannot be ignored. One possible explanation for this case is that GPT-4 cannot completely assert "yes" or "no" on the existence of the causal effect from $x_j$ to $x_i$ with the knowledge obtained in Step 2. In more anthropomorphic terms, this could suggest that GPT-4 sometimes struggles with its decision-making.

For further rigid handling in the application of our method, it is required to prevent the break of "faithfulness," or to extend our quantitative handling of the token generation probability to the one which permits the possibility of this behavior.

**Temperature Dependence of the Fluctuation and the "Faithfulness"**

As described in Eq. (2), probability distribution depends on the temperature parameter. Since higher temperature leads to larger generation probabilities of the token, unexpected at lower temperature, it is possible that this parameter can also have effects on the fluctuation of $p_{ij}$ and the "faithfulness." Although we have fixed this parameter at 0.7 in all the experiments in this work, further discussion on the optimal temperature for this task is expected.

# E   Composition of Adjacency Matrices Representing Causal Structure and Evaluation

## E.1   Composition of Prior Knowledge Matrices

As shown in Algorithm 2, the composition rule of $PK$ depends on the type of SCD method adopted, and the decision criteria of forced and forbidden edges are tentatively set at 0.95 and 0.05, respectively. While PC and DirectLiNGAM can be augmented with constraints for both forced and forbidden directed edges or paths, Exact Search can only be augmented with the constraints for forbidden directed edges. In this section, the composition rule for $PK$ is described in detail for all the SCD algorithms we have adopted in this work.

**For PC**   As for the matrix representation of $PK$, the values of the matrix elements are determined as follows:

- Case 1. If $x_i \to x_j$ is forced (i.e., $p_{ij} \geq 0.95$), then $PK_{ij}$ is set to 1.

- Case 2. If $x_i \to x_j$ is forbidden (i.e., $p_{ij} < 0.05$), then $PK_{ij}$ is set to 0.

- Case 3. If the existence of $x_i \to x_j$ cannot be determined immediately from the domain knowledge generated by GPT-4 (i.e., $0.05 \leq p_{ij} < 0.95$), then $PK_{ij}$ is set to $-1$.

This ternary matrix composition is based on the constraints of the prior knowledge matrix DirectLiNGAM, which will be explained later, to apply the generated $PK$ to DirectLiNGAM as quickly as possible. Although prior knowledge is represented as a matrix in the PC algorithm widely open in "causal-learn," both forced and forbidden edges can be set and the possibility of other unknown edges are explored. This similar properties with DirectLiNGAM means that the prior knowledge for this PC algorithm can be represented in ternary matrix, if we need to do. Therefore, the composition rule of $PK$ for PC is set to be the same as that for DirectLiNGAM in this work, to treat it consistently as possible.

**For DirectLiNGAM**   Although the criteria of setting the values in $PK$ are the same as those for PC, the definition of the value becomes slightly different. Although the prior knowledge for the PC algorithm in the "causal-learn" package corresponds to the existence of directed edges between pairs of variables, the prior knowledge for DirectLiNGAM is determined with the knowledge on directed paths. The values of the matrix elements are determined as below:

- Case 1. If the directed path from $x_i$ to $x_j$ is forced (i.e., $p_{ij} \geq 0.95$), then $PK_{ij}$ is set to 1.

- Case 2. If the directed path from $x_i$ to $x_j$ is forbidden (i.e., $p_{ij} < 0.05$), then $PK_{ij}$ is set to 0.

- Case 3. If the existence of the directed path from $x_i$ to $x_j$ cannot be determined immediately from the domain knowledge generated by GPT-4 (i.e., $0.05 \leq p_{ij} < 0.95$), then $PK_{ij}$ is set to $-1$.

This ternary matrix composition, using 1, 0 and $-1$ is indeed implemented in the software package "LiNGAM."

**For Exact Search**   While $PK$ in cases of PC and DirectLiNGAM is a ternary matrix, on e must be careful that $PK$ in Exact Search is a binary matrix. The values of the matrix elements are determined as below:

- Case 4. If $x_i \to x_j$ is forbidden (i.e., $p_{ij} < 0.05$), then $PK_{ij}$ is set to 0.

- Case 5. If $x_i \to x_j$ is forced, or the existence of this causal relationship cannot be determined immediately from the domain knowledge generated by GPT-4 (i.e., $0.05 \leq p_{ij}$), then $PK_{ij}$ is set to 1.

It must be carefully noted that, although the definition of $PK_{ij} = 0$ in Case 4 for Exact Search is exactly the same as that in Case 2 for PC and DirectLiNGAM, the definition of $PK_{ij} = 1$ in Case 5 for Exact Search encompasses the both Case 1 and Case 3 for PC and DirectLiNGAM. This difference must be taken into account when evaluating $PK$ in comparison with the ground truths, to interpret the results in a unified manner regardless of the SCD methods used.

### E.2 Composition of Ground Truth Matrix

The representation of ground truths in matrix form can be simply realized using a binary matrix, provided that it is determined whether a directed edge exists for every possible pair of variables in the system. The composition rule for the ground truth matrix $\boldsymbol{GT}$ is as follows:

- If $x_j \rightarrow x_j$ exists, then $GT_{ij}$ is set to 1.

- If $x_j \rightarrow x_j$ does not exist, then $GT_{ij}$ is set to 0.

The matrix representations of the ground truth of the benchmark datasets of Auto MPG, DWD, and Sachs shown in Appendix C are expressed as follows:

$$\boldsymbol{GT}_{\text{AutoMPG}} = \begin{pmatrix} 0 & 0 & 0 & 1 & 0 \\ 0 & 0 & 1 & 1 & 0 \\ 1 & 0 & 0 & 0 & 0 \\ 0 & 0 & 0 & 0 & 0 \\ 0 & 0 & 1 & 0 & 0 \end{pmatrix} \tag{10}$$

$$\boldsymbol{GT}_{\text{DWD}} = \begin{pmatrix} 0 & 0 & 0 & 0 & 0 & 0 \\ 1 & 0 & 0 & 1 & 0 & 1 \\ 1 & 0 & 0 & 1 & 0 & 0 \\ 0 & 0 & 0 & 0 & 0 & 0 \\ 1 & 0 & 0 & 0 & 0 & 0 \\ 0 & 0 & 0 & 0 & 0 & 0 \end{pmatrix} \tag{11}$$

$$\boldsymbol{GT}_{\text{Sachs}} = \begin{pmatrix} 0 & 0 & 0 & 0 & 0 & 0 & 0 & 0 & 1 & 0 & 0 \\ 1 & 0 & 0 & 0 & 0 & 0 & 0 & 1 & 1 & 0 & 0 \\ 0 & 0 & 0 & 0 & 1 & 0 & 0 & 0 & 0 & 0 & 0 \\ 0 & 0 & 1 & 0 & 1 & 0 & 0 & 0 & 0 & 0 & 0 \\ 0 & 0 & 0 & 0 & 0 & 0 & 0 & 0 & 0 & 0 & 0 \\ 0 & 1 & 0 & 0 & 0 & 0 & 0 & 1 & 0 & 0 & 0 \\ 0 & 0 & 0 & 0 & 1 & 1 & 0 & 1 & 0 & 0 & 0 \\ 0 & 0 & 0 & 0 & 0 & 0 & 0 & 0 & 1 & 0 & 0 \\ 0 & 0 & 1 & 1 & 0 & 0 & 0 & 0 & 0 & 0 & 0 \\ 0 & 0 & 0 & 0 & 0 & 0 & 0 & 1 & 1 & 0 & 0 \\ 0 & 0 & 0 & 0 & 0 & 0 & 0 & 1 & 1 & 0 & 0 \end{pmatrix} \tag{12}$$

### E.3 Calculation of Metrics for Evaluation of Structural Consistency with Ground Truths (SHD, FPR, FNR, Precision, F1 Score)

Structural metrics such as SHD, FPR, FNR, precision, and F1 score are commonly calculated for performance evaluation in various machine learning and classification contexts, and are compared with the ground truth data. In a similar context, the causal structures inferred by LLM-KBCI and SCD, especially for the benchmark datasets with known ground truths, can also be evaluated using these metrics.

For the practical evaluation of the SCD results in this study, we use the ground truth matrices defined for the benchmark datasets in Eq.(10), (11) and (12) as references, and we measure these metrics using the adjacency matrices that are calculated directly in SCD algorithms or easily transformed from the output causal graphs. Similarly, the calculation of these metrics for the evaluating LLM-KBCI outputs is carried out using $\boldsymbol{PK}$.

However, it must be noted that there can be some arguments on the definition of metrics for $\boldsymbol{PK}$ based on $\boldsymbol{GT}$, because the definition of the matrix elements of $\boldsymbol{PK}$ shown in Appendix E.1 is partially different from that of $\boldsymbol{GT}$ described in Appendix E.2. In particular, although $GT_{ij}$ is a binary variable completely determined with whether $x_j \rightarrow x_i$ exists or not, $PK_{ij}$ can be set to $-1$ for PC and DirectLiNGAM and to 1 for Exact Search. This indicates or includes the case where it is not impossible to definitively assert the presence or absence of $x_j \rightarrow x_i$.

Therefore, although there may be a discussion that a reasonable extension of the definitions of these metrics is required for the case above, in this study, we evaluate these metrics from $|\boldsymbol{PK}|$, in which both $PK_{ij} = 1$ and $PK_{ij} = -1$ are interpreted as a "tentative assertion of the presence of $x_j \rightarrow x_i$" and are treated identically. This processing of $\boldsymbol{PK}$ can also be interpreted as that of temporarily adopting the composition rule of $\boldsymbol{PK}$ for Exact Search as it is for the evaluation of SHD, FPR, FNR, precision, and F1 score, for all SCD methods. With this processing, $\boldsymbol{PK}$ is handled in a unified manner regardless of the SCD methods used. We believe this approach is the best way to maintain the original concept of composing $\boldsymbol{PK}$, while aiming for consistent discussion across all SCD methods.

**Calculation of SHD**   According to the original concept of the structural hamming distance (SHD), this metric is represented as the total number of edge additions, deletions, or reversals that are needed to convert the estimated graph $G'$ into its ground truth graph $G$ (Zheng et al., 2018; Cheng et al., 2022; Hasan et al., 2023). As in our study, if network graphs $G$ and $G'$ are represented by binary matrices $\boldsymbol{G}$ and $\boldsymbol{G'}$, respectively, where all elements are either 0 or 1, then the total number of edge additions ($A$), deletions ($D$), and reversals ($R$) can be simply calculated as follows:

$$A(\boldsymbol{G'}, \boldsymbol{G}) = \sum_{i,j} \mathbf{1}(G_{ij})\mathbf{1}(G_{ji})\mathbf{1}(G'_{ij} - 1) \tag{13}$$

$$D(\boldsymbol{G'}, \boldsymbol{G}) = \sum_{i,j} \mathbf{1}(G'_{ij})\mathbf{1}(G'_{ji})\mathbf{1}(G_{ij} - 1) \tag{14}$$

$$R(\boldsymbol{G'}, \boldsymbol{G}) = \sum_{i,j} \mathbf{1}(G_{ij})\mathbf{1}(G_{ji} - 1)\mathbf{1}(G'_{ij} - 1)\mathbf{1}(G'_{ji}) \tag{15}$$

Here, we introduce the indicator function $\mathbf{1}(x)$, expressed as follows:

$$\mathbf{1}(x) = \begin{cases} 1 & \text{if } x = 0, \\ 0 & \text{otherwise } (x \neq 0). \end{cases} \tag{16}$$

As SHD is defined as $SHD = A + D + R$, it is easily evaluated as follows:

$$SHD(\boldsymbol{G'}, \boldsymbol{G}) = \\ \sum_{i,j} \Big\{ \ \mathbf{1}(G_{ij})\mathbf{1}(G_{ji})\mathbf{1}(G'_{ij} - 1) \ + \ \mathbf{1}(G'_{ij})\mathbf{1}(G'_{ji})\mathbf{1}(G_{ij} - 1) \ + \ \mathbf{1}(G_{ij})\mathbf{1}(G_{ji} - 1)\mathbf{1}(G'_{ij} - 1)\mathbf{1}(G'_{ji}) \ \Big\} \tag{17}$$

For the evaluation of SHD of LLM-KBCI outputs, $SHD(|\boldsymbol{PK}|, \boldsymbol{GT})$ is calculated with Eq. (17).

**Calculation of FPR, FNR, Precision, and F1score**   In the similar context to SHD, for calculation of the metrics such as false positive rate (FPR) and false negative rate (FNR), we prepare the equation for evaluating the number of true positive (TP), false positive (FP), true negative (TN), and false negative (FN) edges as follows:

$$TP(\boldsymbol{G'}, \boldsymbol{G}) = \sum_{i,j} \mathbf{1}(G_{ij} - 1)\mathbf{1}(G'_{ij} - 1) \tag{18}$$

$$FP(\boldsymbol{G'}, \boldsymbol{G}) = \sum_{i,j} \mathbf{1}(G_{ij})\mathbf{1}(G'_{ij} - 1) \tag{19}$$

$$TN(\boldsymbol{G'}, \boldsymbol{G}) = \sum_{i,j} \mathbf{1}(G_{ij})\mathbf{1}(G'_{ij}) \tag{20}$$

$$FN(\boldsymbol{G'}, \boldsymbol{G}) = \sum_{i,j} \mathbf{1}(G_{ij} - 1)\mathbf{1}(G'_{ij}) \tag{21}$$

Then, using Eq. (18)– (21), the definition of FPR, FNR, precision, and F1 score can be expressed as follows:

$$FPR(\boldsymbol{G'}, \boldsymbol{G}) = \frac{FP(\boldsymbol{G'}, \boldsymbol{G})}{TN(\boldsymbol{G'}, \boldsymbol{G}) + FP(\boldsymbol{G'}, \boldsymbol{G})} \tag{22}$$

$$FNR(\boldsymbol{G'}, \boldsymbol{G}) = \frac{FN(\boldsymbol{G'}, \boldsymbol{G})}{TP(\boldsymbol{G'}, \boldsymbol{G}) + FN(\boldsymbol{G'}, \boldsymbol{G})} \tag{23}$$

$$Precision(\boldsymbol{G'}, \boldsymbol{G}) = \frac{TP(\boldsymbol{G'}, \boldsymbol{G})}{TP(\boldsymbol{G'}, \boldsymbol{G}) + FP(\boldsymbol{G'}, \boldsymbol{G})} \tag{24}$$

$$F_1 score(\boldsymbol{G'}, \boldsymbol{G}) = \frac{2\,TP(\boldsymbol{G'}, \boldsymbol{G})}{2\,TP(\boldsymbol{G'}, \boldsymbol{G}) + FN(\boldsymbol{G'}, \boldsymbol{G}) + FP(\boldsymbol{G'}, \boldsymbol{G})} \tag{25}$$

For the evaluation of structural metrics such as FPR of LLM-KBCI outputs, $FPR(|\boldsymbol{PK}|, \boldsymbol{GT})$, $FNR(|\boldsymbol{PK}|, \boldsymbol{GT})$, $Precision(|\boldsymbol{PK}|, \boldsymbol{GT})$ and $F_1 score(|\boldsymbol{PK}|, \boldsymbol{GT})$ are calculated with Eq. (22)–(25).

# F    Algorithm $A$ for Transformation of Cyclic $PK$ into Acyclic Adjacency Matrices and Selection of the Optimal Matrix

As briefly described in Section 3.1, for the case of DirectLiNGAM, acyclicity of $\boldsymbol{PK}$ is also required. Thus, if the $\boldsymbol{PK}$ directly calculated from the probability matrix is cyclic, it must be transformed into an acyclic form. One possible method is to delete the minimum number of edges included in cycles to transform $\boldsymbol{PK}$ into an acyclic matrix. However, it is possible that there are several solutions transformed from the same $\boldsymbol{PK}$ with the minimum manipulation of deleting edges. Therefore, we have decided to carry out causal discovery with DirectLiNGAM for every possible acyclic prior knowledge matrix to select the best acyclic prior knowledge matrix $\boldsymbol{PK}_A$ in terms of statistical modeling. The dataset is again fitted with a structural equation model under the constraint of the causal structure explored with DirectLiNGAM, assuming linear-Gaussian data, and the Bayes Information Criterion (BIC) is calculated. After repeating this process, the acyclic prior knowledge matrix with which the BIC becomes the lowest is selected as $\boldsymbol{PK}_A$.

The overall transformation process is described in Algorithm 3. However, in the practical application of this method, it must be noted that completing the list of acyclic prior knowledge matrices $A$ incurs significant computational costs. Hence, as the number of variables increases, completing this calculation algorithm in a realistic time frame becomes more challenging.

For the future generalization and application of our inference method using DirectLiNGAM, the development of more efficient algorithms for transforming a cyclic matrix into an acyclic one is anticipated.

---

**Algorithm 3** Transformation of Cyclic $\boldsymbol{PK}$ into Acyclic Adjacency Matrices and Selection of the Optimal Matrix

---

    **Input 1:** Cyclic prior knowledge matrix $\boldsymbol{PK_C}$
    **Input 2:** Data $X$ with variables$\{x_1, ..., x_n\}$
    **Input 3:** DirectLiNGAM Algorithm $L(X, \boldsymbol{PK})$
    **Output:** Optimal acyclic matrix $\boldsymbol{PK_A}$
    Initialize the temporal set for matrices $\boldsymbol{T} \leftarrow \{\boldsymbol{PK_C}\}$
    Initialize the temporal set for number of the cycles $\boldsymbol{N} \leftarrow \{\}$
    Initialize the temporal set for acyclic matrices $\boldsymbol{A} \leftarrow \{\}$
    **repeat**
      **for** matrix $\boldsymbol{T_m} \in \boldsymbol{T}$ **do**
        Count the number of cycles in $\boldsymbol{T_m}$ as $N_m$
        Add $N_m$ to $\boldsymbol{N}$
      **end for**
      **if** $\exists \boldsymbol{N_m} \in \boldsymbol{N} \ N_m = 0$ **then**
        Detect all $\boldsymbol{T_m}$, which satisfies $N_m = 0$ and add them to $\boldsymbol{A}$
      **else**
        Initialize the temporal set for modified matrices $\boldsymbol{T'} \leftarrow \{\}$
        **for** $\boldsymbol{T_m} \in \boldsymbol{T}$ **do**
          Initialize the set for edges included in cycles $\boldsymbol{E_m} \leftarrow \{\}$
          Initialize the set for edges to be removed $\boldsymbol{F_m} \leftarrow \{\}$
          Detect all cycles in $\boldsymbol{T_m}$
          For each detected cycle, identify all the edges that form the cycle
          Add these edges to a set of edges to be removed $\boldsymbol{E_m}$
          Detect the most frequent edges "$x_i \leftarrow x_j$" in $\boldsymbol{E_m}$ as $(i, j)_f$
          $\forall (i_f, j_f)$ add $(i_f, j_f)$ to $\boldsymbol{F_m}$
          **for** $(i_f, j_f) \in \boldsymbol{F_m}$ **do**
            $\boldsymbol{T'_m} = \boldsymbol{T_m}$
            $\boldsymbol{T'_m}(i_f, j_f) \leftarrow 0$
            Add $\boldsymbol{T'_m}$ to $\boldsymbol{T'}$
          **end for**
        **end for**
        Replace $\boldsymbol{T}$ with $\boldsymbol{T'}$
      **end if**
    **until** $\boldsymbol{A}$ is not empty
    Initialize the optimal BIC value $B = \text{None}$
    Initialize the optimal acyclic matrix for prior knowledge in DirectLiNGAM $\boldsymbol{A}_{\text{optimal BIC}} = \text{None}$
    **for** $\boldsymbol{A_m} \in \boldsymbol{A}$ **do**
      Calculate adjacency matrix (with the components 0 or 1) of the causal discovery result $\boldsymbol{Adj} = L(X, \boldsymbol{PK} = \boldsymbol{A_m})$
      Fit $X$ with the structural causal equation model represented in $\boldsymbol{Adj}$ assuming linear-Gaussian data
      Calculate BIC with $\boldsymbol{Adj}$ and $X$ as $B_{\text{temp}}$
      **if** $B > B_{\text{temp}}$ **or** $B = \text{None}$ **then**
        $B \leftarrow B_{\text{temp}}$
        $\boldsymbol{A}_{\text{optimal BIC}} = \boldsymbol{A_m}$
      **end if**
    **end for**
    **return** $\boldsymbol{A}_{\text{optimal BIC}}$ as $\boldsymbol{PK_A}$

---

# G   Details of LLM-KBCI Results

It is also valuable to examine the details of the probability matrices generated by LLM-KBCI, both for the basic discussion on whether LLMs can generate a valid interpretation of causality from a domain expert's point of view, and for understanding the characteristics of SCP. In this section, the probability matrices generated by GPT-4 for Auto MPG data and DWD climate data, which are relatively easy to interpret within common daily knowledge, are shown and briefly interpreted. For comparison among various SCP patterns (Patterns 1–4) using the same SCD method as much as possible, the probability matrices generated by GPT-4 with SCP are shown only for DirectLiNGAM. We also briefly present the probability matrices of LLM-KBCI for the sampled sub-dataset of health screening results.

## G.1   LLM-KBCI for Auto MPG data

In Table 11, the probabilities of causal relationships of pairs of variables in Auto MPG data are shown. The cells highlighted in green are the ones in which the directed edges are expected to appear from the ground truths shown in Figure 3.

For all the prompting patterns, although the probability of "Weight"→"Displacement", which is interpreted as one of the ground truth directed edges, is 0, the probability of reversed edge "Displacement"→"Weight" is non-zero and over 0.95 in Patterns 1–4. For understanding this behavior and elucidating the true causal relationship between these two variables, further discussion is required, including the possibility of the hidden common causes that are excluded from the dataset we have used.

In addition to that, although we do not believe the existence of the directed edge of "Displacement"→"Acceleration," the probability of this causal relationship is over 0.85 for all the prompting patterns. This may be due to the property of the prompting for evaluating the probability. As shown in Table 2, GPT-4 is allowed to judge the existence of both direct and indirect causal relationships, to acquire a positive answer even if any intervening variables are not included in the dataset. However, for example, considering that the probabilities of both "Displacement"→"Horsepower" and "Horsepower"→"Acceleration," which are part of the ground truths, are relatively high, it is also possible that GPT-4 supports the hypothesis of some impact from "Displacement" on "Acceleration" partially due to the confidence in the indirect causal relationship of "Displacement"→"Horsepower"→"Acceleration". If one wants to distinguish the direct and indirect causal relationships in the interpretation of the probability matrix, investigation of the response from LLMs for the first prompting may lead to further understanding.

Some differences that can be related to the prompting patterns can also be observed. For example, the probability of "Horsepower"→"Mpg" in Pattern 1 is much smaller than other patterns. Moreover, the probabilities of "Horsepower"→"Acceleration" in Patterns 1 and 3 are smaller than other patterns, in which the probability of this edge is almost 1. A possible explanation of these behaviors is that the decision-making of GPT-4 is unsettled with SCP, in which the causal structure inferred by DirectLiNGAM shown in Figure 4 (c) is included. As neither "Horsepower"→"Acceleration" nor "Horsepower"→"Mpg" appears in Figure 4 (c), despite the confidence in the existence of these edges only from the domain knowledge, the decision-making of GPT-4 may become more careful, taking into account the result of SCD. It is desired to elucidate what kinds of decision-making of LLMs are likely to be affected by SCP in future work.

## G.2   LLM-KBCI for DWD climate data

In Table 12, the probabilities of the causal relationships of pairs of variables in DWD climate data are shown. The cells highlighted in green are the ones in which the directed edges are expected to appear from the ground truths shown in Figure 5.

For all the prompting patterns, it is confirmed that all of the probabilities of the causal effects on "Altitude," "Longitude," and "Latitude" from other variables are 0. As these three variables are geographically given and fixed, the interpretation by GPT-4 that they act as parent variables that are not influenced by other factors is completely reasonable. Although "Altitude" and "Latitude" are somehow influenced according to the result of DirectLiNGAM without prior knowledge as shown in Figure 6 (c), SCP including these unnatural results

Table 11: Probabilities of the causal relationships suggested by GPT-4 in Auto MPG data. The cells in which the directed edges are expected to appear from the ground truths as shown in Figure 3 are highlighted in green.

**Pattern 0**

| EFFECTED\CAUSE | "Displacement" | "Mpg" | "Horsepower" | "Weight" | "Acceleration" |
|---|---|---|---|---|---|
| "Displacement" | - | 0.000 | 0.000 | 0.000 | 0.000 |
| "Mpg" | 0.999 | - | 0.997 | 1.000 | 1.000 |
| "Horsepower" | 0.999 | 0.000 | - | 0.000 | 0.000 |
| "Weight" | 0.635 | 0.000 | 0.000 | - | 0.000 |
| "Acceleration" | 0.996 | 0.023 | 0.998 | 0.998 | - |

**Pattern 1**

| EFFECTED\CAUSE | "Displacement" | "Mpg" | "Horsepower" | "Weight" | "Acceleration" |
|---|---|---|---|---|---|
| "Displacement" | - | 0.000 | 0.000 | 0.000 | 0.000 |
| "Mpg" | 1.000 | - | 0.128 | 0.484 | 0.058 |
| "Horsepower" | 1.000 | 0.056 | - | 0.001 | 0.000 |
| "Weight" | 0.994 | 0.000 | 0.000 | - | 0.000 |
| "Acceleration" | 0.859 | 0.000 | 0.828 | 0.998 | - |

**Pattern 2**

| EFFECTED\CAUSE | "Displacement" | "Mpg" | "Horsepower" | "Weight" | "Acceleration" |
|---|---|---|---|---|---|
| "Displacement" | - | 0.000 | 0.000 | 0.000 | 0.000 |
| "Mpg" | 1.000 | - | 0.999 | 1.000 | 0.984 |
| "Horsepower" | 1.000 | 0.000 | - | 0.000 | 0.000 |
| "Weight" | 0.997 | 0.000 | 0.000 | - | 0.000 |
| "Acceleration" | 0.995 | 0.002 | 0.996 | 0.999 | - |

**Pattern 3**

| EFFECTED\CAUSE | "Displacement" | "Mpg" | "Horsepower" | "Weight" | "Acceleration" |
|---|---|---|---|---|---|
| "Displacement" | - | 0.000 | 0.000 | 0.000 | 0.000 |
| "Mpg" | 0.977 | - | 0.969 | 0.754 | 0.547 |
| "Horsepower" | 1.000 | 0.051 | - | 0.696 | 0.010 |
| "Weight" | 0.954 | 0.000 | 0.000 | - | 0.000 |
| "Acceleration" | 0.981 | 0.000 | 0.435 | 0.809 | - |

**Pattern 4**

| EFFECTED\CAUSE | "Displacement" | "Mpg" | "Horsepower" | "Weight" | "Acceleration" |
|---|---|---|---|---|---|
| "Displacement" | - | 0.000 | 0.000 | 0.000 | 0.000 |
| "Mpg" | 0.995 | - | 0.994 | 0.997 | 0.940 |
| "Horsepower" | 0.999 | 0.314 | - | 0.006 | 0.000 |
| "Weight" | 0.999 | 0.000 | 0.012 | - | 0.000 |
| "Acceleration" | 0.964 | 0.000 | 0.989 | 0.814 | - |

has not affected the decision-making by GPT-4. From this behavior, it is inferred that the response regarding axiomatic and self-evident matters from GPT-4 is robust and not likely to be affected by SCP, even if the SCD result exhibits obviously unnatural behaviors.

In addition, while "Longitude"→"Temperature," which are assumed to be a ground truth, is not likely to be asserted by GPT-4, "Temperature"→"Precipitation," which is not expected to be a ground truth, is likely to be asserted by GPT-4, across all the prompting patterns. For further interpretation of these unexpected behaviors from our ground truths, investigation of the response generated in the first prompting process is recommended. It is also interesting that although the probabilities of "Longitude"→"Precipitation" are 0 in Patterns 0–2, they become non-zero finite values in Patterns 3 and 4, in which the causal coefficient of this

directed edge calculated with DirectLiNGAM is included in SCP. This behavior may be a glimpse that SCP can assist the decision-making of GPT-4 even if it generates an incomplete response on causal relationships with its background knowledge.

Table 12: Probabilities of the causal relationships suggested by GPT-4 in DWD climate data. The cells in which the directed edges are expected to appear from the ground truths as shown in Figure 5 are highlighted in green.

Pattern 0

| EFFECTED\CAUSE | "Altitude" | "Temperature" | "Precipitation" | "Longitude" | "Sunshine" | "Latitude" |
|---|---|---|---|---|---|---|
| "Altitude" | - | 0.000 | 0.000 | 0.000 | 0.000 | 0.000 |
| "Temperature" | 1.000 | - | 0.891 | 0.000 | 1.000 | 1.000 |
| "Precipitation" | 1.000 | 0.999 | - | 0.000 | 0.001 | 0.995 |
| "Longitude" | 0.000 | 0.000 | 0.000 | - | 0.000 | 0.000 |
| "Sunshine" | 1.000 | 0.000 | 0.998 | 0.000 | - | 1.000 |
| "Latitude" | 0.000 | 0.000 | 0.000 | 0.000 | 0.000 | - |

Pattern 1

| EFFECTED\CAUSE | "Altitude" | "Temperature" | "Precipitation" | "Longitude" | "Sunshine" | "Latitude" |
|---|---|---|---|---|---|---|
| "Altitude" | - | 0.000 | 0.000 | 0.000 | 0.000 | 0.000 |
| "Temperature" | 0.384 | - | 0.034 | 0.000 | 0.856 | 0.011 |
| "Precipitation" | 0.025 | 0.999 | - | 0.000 | 0.036 | 0.026 |
| "Longitude" | 0.000 | 0.000 | 0.000 | - | 0.000 | 0.000 |
| "Sunshine" | 0.006 | 0.011 | 0.008 | 0.000 | - | 0.596 |
| "Latitude" | 0.000 | 0.000 | 0.000 | 0.000 | 0.000 | - |

Pattern 2

| EFFECTED\CAUSE | "Altitude" | "Temperature" | "Precipitation" | "Longitude" | "Sunshine" | "Latitude" |
|---|---|---|---|---|---|---|
| "Altitude" | - | 0.000 | 0.000 | 0.000 | 0.000 | 0.000 |
| "Temperature" | 0.997 | - | 0.007 | 0.000 | 0.999 | 0.989 |
| "Precipitation" | 0.739 | 0.999 | - | 0.000 | 0.000 | 0.384 |
| "Longitude" | 0.000 | 0.000 | 0.000 | - | 0.000 | 0.000 |
| "Sunshine" | 0.874 | 0.010 | 0.976 | 0.000 | - | 0.981 |
| "Latitude" | 0.000 | 0.000 | 0.000 | 0.000 | 0.000 | - |

Pattern 3

| EFFECTED\CAUSE | "Altitude" | "Temperature" | "Precipitation" | "Longitude" | "Sunshine" | "Latitude" |
|---|---|---|---|---|---|---|
| "Altitude" | - | 0.000 | 0.000 | 0.000 | 0.000 | 0.000 |
| "Temperature" | 0.919 | - | 0.016 | 0.003 | 0.615 | 0.973 |
| "Precipitation" | 0.585 | 0.996 | - | 0.175 | 0.002 | 0.008 |
| "Longitude" | 0.000 | 0.000 | 0.000 | - | 0.000 | 0.000 |
| "Sunshine" | 0.039 | 0.000 | 0.001 | 0.875 | - | 0.199 |
| "Latitude" | 0.000 | 0.000 | 0.000 | 0.000 | 0.000 | - |

Pattern 4

| EFFECTED\CAUSE | "Altitude" | "Temperature" | "Precipitation" | "Longitude" | "Sunshine" | "Latitude" |
|---|---|---|---|---|---|---|
| "Altitude" | - | 0.000 | 0.000 | 0.000 | 0.000 | 0.000 |
| "Temperature" | 0.982 | - | 0.023 | 0.029 | 0.990 | 0.958 |
| "Precipitation" | 0.826 | 0.987 | - | 0.927 | 0.010 | 0.797 |
| "Longitude" | 0.000 | 0.000 | 0.000 | - | 0.000 | 0.000 |
| "Sunshine" | 0.534 | 0.021 | 0.387 | 0.013 | - | 0.638 |
| "Latitude" | 0.000 | 0.000 | 0.000 | 0.000 | 0.000 | - |

### G.3 LLM-KBCI for Dataset of Health Screening Results

In Table 13, the probabilities of the causal relationships of pairs of variables in our sampled sub-dataset of health screening results are shown. The cells highlighted in red are the ones in which the directed edges are expected to appear as described in Appendix C.4. In contrast, since "Age" is an unmodifiable background factor, it can be concluded that it is not a descendant of any other variables. Therefore, the probabilities in the cells highlighted in blue are expected to be 0.

Across all the prompting patterns, it is confirmed that all of probabilities of the causal effects on "Age" from other variables are indeed 0. From this fact, it is likely to be regarded by GPT-4 as axiomatic and self-evident that "Age" cannot be affected from other variables, and the judge of the causal relationships is not influenced by SCP, even if the SCD result exhibits obviously unnatural behaviors as shown in Figure 10.

Table 13: Probabilities of the causal relationships suggested by GPT-4 in the sampled sub-dataset of health screening results. The cells in which the directed edges are expected to appear from the ground truths are highlighted in red. In contrast, the probabilities in the cells highlighted in blue, are expected to be zero, since "Age" is expected to be a parent variable for all other variables.

**Pattern 0**

| EFFECTED\CAUSE | "BMI" | "Waist" | "SBP" | "DBP" | "HbA1c" | "LDL" | "Age" |
|---|---|---|---|---|---|---|---|
| "BMI" | - | 0.994 | 0.000 | 0.000 | 0.000 | 0.000 | 0.901 |
| "Waist" | 1.000 | - | 0.000 | 0.000 | 0.000 | 0.000 | 0.353 |
| "SBP" | 0.999 | 0.962 | - | 0.998 | 0.987 | 0.000 | 0.626 |
| "DBP" | 0.998 | 0.995 | 0.993 | - | 0.000 | 0.000 | 0.001 |
| "HbA1c" | 0.998 | 0.998 | 0.000 | 0.000 | - | 0.000 | 0.986 |
| "LDL" | 0.988 | 0.967 | 0.000 | 0.000 | 0.000 | - | 0.002 |
| "Age" | 0.000 | 0.000 | 0.000 | 0.000 | 0.000 | 0.000 | - |

**Pattern 1**

| EFFECTED\CAUSE | "BMI" | "Waist" | "SBP" | "DBP" | "HbA1c" | "LDL" | "Age" |
|---|---|---|---|---|---|---|---|
| "BMI" | - | 0.312 | 0.000 | 0.000 | 0.014 | 0.000 | 0.076 |
| "Waist" | 1.000 | - | 0.000 | 0.000 | 0.023 | 0.000 | 0.043 |
| "SBP" | 0.999 | 0.912 | - | 0.999 | 0.997 | 0.000 | 0.302 |
| "DBP" | 0.998 | 0.421 | 0.050 | - | 0.000 | 0.000 | 0.019 |
| "HbA1c" | 0.517 | 0.503 | 0.101 | 0.000 | - | 0.000 | 0.170 |
| "LDL" | 0.008 | 0.527 | 0.000 | 0.000 | 0.000 | - | 0.517 |
| "Age" | 0.000 | 0.000 | 0.000 | 0.000 | 0.000 | 0.000 | - |

**Pattern 2**

| EFFECTED\CAUSE | "BMI" | "Waist" | "SBP" | "DBP" | "HbA1c" | "LDL" | "Age" |
|---|---|---|---|---|---|---|---|
| "BMI" | - | 0.998 | 0.001 | 0.001 | 0.996 | 0.000 | 0.093 |
| "Waist" | 0.999 | - | 0.000 | 0.003 | 0.959 | 0.007 | 0.099 |
| "SBP" | 0.998 | 0.994 | - | 0.983 | 0.994 | 0.040 | 0.207 |
| "DBP" | 0.997 | 0.975 | 0.984 | - | 0.983 | 0.002 | 0.115 |
| "HbA1c" | 0.982 | 0.608 | 0.002 | 0.000 | - | 0.000 | 0.723 |
| "LDL" | 0.994 | 0.946 | 0.000 | 0.000 | 0.452 | - | 0.171 |
| "Age" | 0.000 | 0.000 | 0.000 | 0.000 | 0.000 | 0.000 | - |

**Pattern 3**

| EFFECTED\CAUSE | "BMI" | "Waist" | "SBP" | "DBP" | "HbA1c" | "LDL" | "Age" |
|---|---|---|---|---|---|---|---|
| "BMI" | - | 0.003 | 0.000 | 0.000 | 0.868 | 0.923 | 0.306 |
| "Waist" | 1.000 | - | 0.000 | 0.000 | 0.983 | 0.000 | 0.076 |
| "SBP" | 1.000 | 0.855 | - | 0.959 | 0.999 | 0.000 | 0.235 |
| "DBP" | 1.000 | 0.032 | 0.140 | - | 0.021 | 0.000 | 0.095 |
| "HbA1c" | 0.967 | 0.634 | 0.000 | 0.000 | - | 0.000 | 0.046 |
| "LDL" | 0.562 | 0.165 | 0.000 | 0.000 | 0.085 | - | 0.013 |
| "Age" | 0.000 | 0.000 | 0.000 | 0.000 | 0.000 | 0.000 | - |

**Pattern 4**

| EFFECTED\CAUSE | "BMI" | "Waist" | "SBP" | "DBP" | "HbA1c" | "LDL" | "Age" |
|---|---|---|---|---|---|---|---|
| "BMI" | - | 0.993 | 0.000 | 0.000 | 0.024 | 0.006 | 0.037 |
| "Waist" | 1.000 | - | 0.000 | 0.000 | 0.957 | 0.000 | 0.395 |
| "SBP" | 0.999 | 0.982 | - | 0.001 | 0.998 | 0.000 | 0.795 |
| "DBP" | 0.994 | 0.204 | 0.985 | - | 0.408 | 0.000 | 0.926 |
| "HbA1c" | 0.824 | 0.391 | 0.000 | 0.000 | - | 0.000 | 0.176 |
| "LDL" | 0.485 | 0.403 | 0.000 | 0.000 | 0.000 | - | 0.027 |
| "Age" | 0.000 | 0.000 | 0.000 | 0.000 | 0.000 | 0.000 | - |

