# OpenReview forum: "Integrating Large Language Models in Causal Discovery: A Statistical Causal Approach"
_TMLR — Rejected by TMLR_

### Review · Reviewer_SVi9 · 2024-06-05

**Summary Of Contributions:**

This paper considered LLM as an expert prior in the **data-driven** causal discovery. Specifically, LLM is set as a set of queries to analyze the plausibility of the causal discovery results. The discovered results seem more human compatible, compared with traditional methods.

**Audience:**

Yes

**Broader Impact Concerns:**

See the request changes

**Claims And Evidence:**

No

**Requested Changes:**

**About conceptual contribution**

I do agree this paper has sort of interesting insights. However, I have concerns about such methods in the real-world causal discovery such as healthcare. Indeed, the LLM is often trained as a black-box and lacks a clear underlying mechanism. Therefore the expert prior should always have certain bias or misspecification in certain questions. This is crucial in practical causal discovery.

For example, in Figure 1, the X-ray to cancer question has certain ambiguity in a certain sense. Indeed we can not causally infer if someone has cancer, the X-ray will be True. Alternatively we can infer this relation in probability. I.e, if someone has cancer, the probability of X-ray=Ture is 95%. This is a much more reasonable argument. If we take a closer look at this example, prior knowledge provides a better bayes probability rather than the true causal probability. Therefore I think the main contribution should not simply be causal discovery.

Another concern lies in Step 3, the probability estimation from GPT. I also have concerns about creating such probabilistic outputs. As far as I understand, this paper adopted Monte Carlo estimation to measure the probability. Is there any alternative to do so? I feel like current probability estimation seems a bit awkward.

**Suggestions**

I think this paper itself is worthy of publication. However, current submissions lack rigors in the context of causal discovery. I carefully checked the limitation sections and felt rather limited. I would like this paper could have an effective revision based on
- Clearly illustrate the limitations and risks of using current LLM in data-driven causal discovery.
- Showing the failure scenarios that LLM worsens the causal discovery.
- Clearly showing how the probability of confidence is obtained, including its sensitivity and reliability.

**Strengths And Weaknesses:**

**Pros**

Integrating proper expert’s knowledge into the causal discovery is quite interesting and promising in the discovery science. Indeed there is a long term belief that human plausible prior could regularize the causal discovery. This paper interestingly considered this and leveraged LLM as a proxy of the human. I feel like this direction is worthy, promising and exciting.

**Cons**

In general, my concerns are mainly conceptual. I feel like causal-discovery may not be a proper name for these kinds of methods. Indeed, I feel like it’s more or less Bayes structure discovery. Please see the request changes for my concerns.

---

> ### Author Response · Authors · 2024-07-16
> **Answer to Reviewer SVi9**
>
> We sincerely appreciate the insightful comments and constructive criticism you have provided. We have uploaded the revised version, in which the revised parts are written in red font. This manuscript includes the revisions in accord with the requested changes you have provided us with. The details are as follows:
>
> >1. Clearly illustrate the limitations and risks of using current LLM in data-driven causal discovery.
>
> >2. Showing the failure scenarios that LLM worsens the causal discovery.
>
> We have updated the subsection "Limitations of This Work" in Section 5. As indicated in red font, we have made a thorough effort to illustrate the limitations of using current LLMs in our method and to show the risks of LLMs worsening the results of SCD.
>
>
> >3. Clearly showing how the probability of confidence is obtained, including its sensitivity and reliability.
>
> We have added explanations and discussions on this topic in Appendix D. Here, we provide a detailed explanation of how the token generation probability is used as a measure of confidence in the causal relationships. Additionally, we have conducted a sensitivity analysis to discuss the reliability of the decision of the prior knowledge matrix.

---

### Review · Reviewer_1vYs · 2024-06-30

**Summary Of Contributions:**

The paper introduces an approach that combines statistical causal discovery (SCD) with knowledge-based causal inference (KBCI) using large language models (LLMs). The authors present statistical causal prompting (SCP) method, which employs the domain expertise embedded in LLMs to improve the accuracy of causal models derived from SCD algorithms. The paper validates this method through experiments on benchmark datasets and an unpublished real-world dataset, demonstrating significant enhancements in SCD results when supplemented with LLM-provided background knowledge. Notably, these improvements are achieved even when the dataset is not part of the LLM's training data.

**Audience:**

Yes

**Broader Impact Concerns:**

1.  The potential for the LLM to inherit and propagate biases present in its training data, and how this might affect causal inference results.

2.  The implications of using such a system in high-stakes domains like healthcare or economics, where incorrect causal inferences could lead to significant consequences.

3.  A discussion on the transparency and interpretability of the models produced by the integration of LLMs and SCD, and the potential for "black-box" decision-making.

**Claims And Evidence:**

Yes

**Requested Changes:**

1. The authors should discuss the potential impact of using different LLMs and whether the choice of GPT-4 (e.g., smaller models) introduces any biases or limitations to the findings. And also have a comparison and discussion between using probability critic and natural language critic for evaluation of causal relationship.

2. Including a sensitivity analysis to understand how variations in the LLM's confidence levels affect causal inference outcomes would strengthen the study.

3. The paper should address the potential for overfitting when using SCP, particularly with smaller datasets or those with inherent biases.

**Strengths And Weaknesses:**

* Strengths

1. The integration of SCD and LLM-KBCI through SCP is a significant advancement in causal discovery, enhancing model accuracy.

2. The detailed explanation of the algorithms and prompting techniques used provides clarity and replicability to the study.

* Weaknesses

1. While the results are promising, the generalization across different types of datasets and LLMs needs further exploration to establish broader applicability.

2.  The paper acknowledges that the transformation of cyclic matrices into acyclic ones for DirectLiNGAM is computationally expensive, which could be a barrier for some applications.

---

> ### Author Response · Authors · 2024-07-16
> **Answer to Reviewer 1vYs**
>
> We sincerely appreciate the fruitful comments and important criticisms on the broader impact concerns you have provided. We have uploaded the revised version, in which the revised parts are written in red font. This manuscript includes the revisions in accord with the requested changes and broader impact concerns you have provided us with. The details are as follows:
>
> For Requested Changes:
>
> >1. The authors should discuss the potential impact of using different LLMs and whether the choice of GPT-4 (e.g., smaller models) introduces any biases or limitations to the findings. And also have a comparison and discussion between using probability critic and natural language critic for evaluation of causal relationship.
>
> In the subsection "Limitations of This Work" in Section 5, we have appended in red font additional discussions on adopting other LLMs and the importance of further elaboration to prevent hallucinations due to biases in the training data.
> Regarding the critics for causal relationship evaluation, we recognized that these topics are briefly introduced in Sections 1 and 2. However, we intentionally avoid deepening these philosophical discussions on causality, as they are beyond the scope of our research. We have clearly stated that the aim of this work is to improve the performance of SCD and LLM-KBCI through the integration of these approaches.
>
>
> >2. Including a sensitivity analysis to understand how variations in the LLM's confidence levels affect causal inference outcomes would strengthen the study.
>
> We have added a supplemental section on this topic as Appendix D. Here, to clarify how the LLM’s confidence levels can affect the decision of the prior knowledge matrix, we provide a detailed analysis on sensitivity, evaluating the standard errors of the mean probabilities of confidence.
>
>
> >3. The paper should address the potential for overfitting when using SCP, particularly with smaller datasets or those with inherent biases.
>
> We have updated the subsection "Limitations of This Work" in Section 5 to include a discussion on the potential for overfitting in SCP, highlighted in red font. Since we have demonstrated the effectiveness of our method on health screening datasets containing bias, as described in Section 4.2, we have also mentioned this again to emphasize the importance of robustness for small or biased datasets.
>
>
> For Broader Impact Concerns:
>
> >1. The potential for the LLM to inherit and propagate biases present in its training data, and how this might affect causal inference results.
>
> In the subsection "Limitations of This Work" in Section 5, we have added the discussions on this topic in red font, emphasizing the need for further elaboration to prevent hallucinations due to biases in the training data.
>
>
> >2. The implications of using such a system in high-stakes domains like healthcare or economics, where incorrect causal inferences could lead to significant consequences.
>
> >3. A discussion on the transparency and interpretability of the models produced by the integration of LLMs and SCD, and the potential for "black-box" decision-making.
>
> We have added in red font a discussion covering these topics in the subsection “Broader Impact Statement” in Section 5.

---

### Review · Reviewer_3L5q · 2024-07-10

**Summary Of Contributions:**

The authors introduce a prompting strategy for causal discovery. The intuition is to first use a statistical causal discovery method, pass the result in some form to an LLM to find corrections (using a self-criticising prompting strategy), and rerunning the statistical method with the results from the LLM.

The authors evaluate their method on 3 datasets (all with a small number of variables), and a new dataset which will not be disclosed.

**Audience:**

Yes

**Claims And Evidence:**

No

**Requested Changes:**

- The paper would be significantly stronger if the reproducibility is increased through public code and evaluation on an open source model.
- Under related work it is mentioned that the held-out dataset helps motivate the use of existing work. However in the experimental section it is not clear when there is a comparison to this work.
- I did not understand the need to for measuring log probabilities multiple times. How important is using M=5 for these experiments?
- Is it correct that no in-context learning is used in the prompt?
- It is unclear what are 'forbidden or forced causal relationships' in this context.
- In my experience very few probabilities fall in [0.05, 0.95], so doesn't this hyperparameter setting enforce everything to be either 'forbidden or forced'?
- There are a lot of metrics in table 3. Without the relevant background knowledge it is hard to understand what is going on and what each means.
- If I understand correctly, there is no standard deviation on the result, yet this could be quite high given the randomness of the GPT model with temperature 0.7. This should be mentioned, especially since the relative results on the different patterns seem quite random.
- I did not understand (quickly, at least) what table 4 A was trying to show.

Minor textual changes:
- Abstract: The sentence starting with 'Experiments have revealed' is hard to parse
- Introduction: The claim that "LLMs can be expected to perform objective evaluation of causal relationships" is strong and should be motivated with a thorough study.
- Introduction: "observed in **inference themes** involving". Unclear what is meant here.
- 1.2: space extra after LLM inference
- Footnote 3: "Hidden common causes = hidden confounder?"
- 3.1: notations -> notation, the the input elements
- 4.1: 'the baseline result is **that wo**"
- Table 3: the the
- Table 3: I suppose blue highlights the best result, however in the CFI and RMSEA columns often Baseline A is strongest but not marked.
- Table 3: Is the number in the parantheses (eg 5 (8) ) meant to be the standard deviation? I suppose not, but it's not clear what it is meant to represent.
- Figure 2b: Is this the output of pattern 2 or 4?

**Strengths And Weaknesses:**

Strengths:
- Combining traditional causal discovery methods with LLMs for prior knowledge is an interesting idea.
- Quite a few variants of the method are tried, although there is no conclusive evidence for either.
- Using a held-out dataset helps motivate the generality of the method

Weaknesses:
- The method is not analysed from a theoretical point of view. It is unclear how the LLM could impact traditional issues in causal discovery, or if there are any guarantees on that front.
- It is unclear how or if this method would scale to problems with more variables.
- The reproducibility of the method is low. It is unclear if code will be provided, the fourth dataset will not be shared and GPT-4 is used, which changes versions on a whim and is closed source (in fact the version used in this paper is already offline).
- The paper mentions several related works also exploring the integration of LLMs and causal discovery, but does not experimentally compare to them or analyse why or when this method would be preferred. The extra changes focusing on quantitative properties in the prompting don't seem to have a major effect experimentally. I am not familiar with this related work and cannot judge the novelty.

---

> ### Author Response · Authors · 2024-07-16
> **Answer to Reviewer 3L5q (part 1)**
>
> We sincerely appreciate your careful reading and your fruitful comments. We have uploaded the revised version, in which the revised parts are written in red font. This manuscript includes the revisions in accord with the requested changes you have provided us with. The details are as follows:
>
> >1. The paper would be significantly stronger if the reproducibility is increased through public code and evaluation on an open source model.
>
> We have already uploaded the zip-file of the supplemental materials, which includes the code for this work, to OpenReview. Furthermore, we are ready to append the link to our GitHub page, which includes the code, after this paper is accepted. We intentionally avoid writing the link to maintain the double-blind review process.
> Moreover, although we recognize the importance of experiments on other LLMs, we have intentionally fixed the LLM to GPT-4 to maintain consistency and control across the trials, as mentioned in Footnote 4. Our goal in this work is to explore the effectiveness of integrating LLMs into SCD via SCP from various SCP patterns. Instead, in the subsection “Limitations of This Work” in Section 5, we have added a comment on the importance of verifying the universal effectiveness of the proposed method across different LLMs.
>
>
> >2. Under related work it is mentioned that the held-out dataset helps motivate the use of existing work. However in the experimental section it is not clear when there is a comparison to this work.
>
> In the initial version, we intended to convey that since our experiments demonstrated the effectiveness of integrating LLMs into SCD, both on open benchmark datasets and on an unpublished dataset, this fact can also support the validity of other existing works on LLM-guided causal inference. To express this clearly, we have revised that part in red font in Section 2, and we have also added this discussion in Section 4.2.
>
>
> >3. I did not understand the need to for measuring log probabilities multiple times. How important is using M=5 for these experiments?
>
> As explained in Section 3.1, there is a slight fluctuation in the log probability output from GPT-4, so we have adopted the mean value of the probability measured multiple times. The multiple measurements are important not only for confirming reproducibility and obtaining reliable values, but also for evaluating the fluctuation and sensitivity of our method, which are discussed in Appendix D. Although we have shown the experimental setting M=5 for this work, there is no necessity for M to be exactly 5. We chose this value considering the budget and time resources, as there are limitations on the usage and rate of using the OpenAI API.
>
>
> >4. Is it correct that no in-context learning is used in the prompt?
>
> Although in Pattern 0 in-context learning is not used, the prompts include the SCD result in Patterns 1 - 4. In this sense, in-context learning is adopted in Patterns 1 - 4. To clarify this fact, we have added this technical explanation in red font in Sections 1.2 and 3.1.
>
>
> >5. It is unclear what are 'forbidden or forced causal relationships' in this context.
>
> We have added in red font the explanation for the meaning of “forbidden” and “forced” in Section 3.1.
>
>
> >6. In my experience very few probabilities fall in [0.05, 0.95], so doesn't this hyperparameter setting enforce everything to be either 'forbidden or forced'?
>
> We have shown the matrices of the mean probabilities for the AutoMPG, DWD, and health-screening datasets in Appendix G. Although many probabilities fall outside [0.05, 0.95], a notable number do fall within this range. We interpret that the existence of causal relationships whose probabilities are within [0.05, 0.95] indicates that these relationships are not determined solely from domain knowledge. This behavior allows for a degree of freedom in the SCD process, where data-driven causal discovery can help determine the most likely causal relationships.
>
>
> >7. There are a lot of metrics in table 3. Without the relevant background knowledge it is hard to understand what is going on and what each means.
>
> We have added in red font the explanation for the context of using these metrics in Section 4.1 and in the caption of Table 3. Furthermore, for the readers who want to understand the detailed definitions of these metrics, we have explained SHD, FPR, FNR, precision, and F1 score in Appendix E. We have also added in the main body the reference for the original works discussing CFI, RMSEA, and BIC.
>
> >8. If I understand correctly, there is no standard deviation on the result, yet this could be quite high given the randomness of the GPT model with temperature 0.7. This should be mentioned, especially since the relative results on the different patterns seem quite random.
>
> We have added a supplemental section on this topic as Appendix D. In this section, we have evaluated the fluctuation as standard errors to assess the sensitivity and reliability of our method.

---

> ### Author Response · Authors · 2024-07-16
> **Answer to Reviewer 3L5q (part 2)**
>
> >9. I did not understand (quickly, at least) what table 4 A was trying to show.
>
> We have added in red font an explanation for interpreting the results shown in Table 4A, in the caption of Table 4.
>
>
> For Minor textual changes:
>
> >10. Abstract: The sentence starting with 'Experiments have revealed' is hard to parse
>
> >11. Introduction: The claim that "LLMs can be expected to perform objective evaluation of causal relationships" is strong and should be motivated with a thorough study.
>
> >12. Introduction: "observed in inference themes involving". Unclear what is meant here.
>
> >13. 1.2: space extra after LLM inference
>
> >14. Footnote 3: "Hidden common causes = hidden confounder?"
>
> >15. 3.1: notations -> notation, the the input elements
>
> >16. 4.1: 'the baseline result is that wo"
>
> >17. Table 3: the the
>
> We have reflected your comments above in red font in the revised manuscript.
>
>
> >18. Table 3: I suppose blue highlights the best result, however in the CFI and RMSEA columns often Baseline A is strongest but not marked.
>
> The blue highlights indicate the best result among Patterns 0—4, excluding Baseline A, to try to explore the optimal prompting pattern for each dataset and SCD algorithm. To clarify this intention, we have revised the caption of Table 3.
>
>
> >19. Table 3: Is the number in the parantheses (eg 5 (8) ) meant to be the standard deviation? I suppose not, but it's not clear what it is meant to represent.
>
> The numbers in parentheses indicate not the standard deviations but SHD, FPR, FNR, precision, and F1score of PK with ground truths for evaluating the performance of LLM-KBCI. To state this clearly, we have added this explanation in red font in the caption of Table 3.
>
>
> >20. Figure 2b: Is this the output of pattern 2 or 4?
>
> Fig. 2(b) shows the output from DirectLiNGAM, which yields the same causal models for both Patterns 2 and 4.

---

### Author Response · Authors · 2024-07-16
**Revision with Requested Changes Complete**

Dear reviewers,

We really appreciate again your the valuable feedback!

The revised paper has been uploaded, and the revised parts are written in red font.

We look forward to further discussion.

Best regards, Authors of Paper 2730

---

### Decision · Action_Editor_8bgj · 2024-08-05

**Recommendation:** Reject

**Comment:**

Two reviewers out of three recommended the paper to be accepted, even if their recommendation il "leaning accept", while the third reviewer recommended rejection.
I went through all reviews, rebuttals from authors and found that the vast majority of issues raised by the reviewers were addressed and solved in a sensible manner, as two out of three reviewers agree upon. I also put particular attention to the review and the argument from the third reviewer who recommended rejection. I found the arguments there developed to make sense and being useful to foster the discussion. If what suggested by SVi9 is achieved can it will significantly improve the impact of the interesting work of the authors, exactly as SVi9 agrees upon in the last comments to the recommendation,
I also think the approach should be made more specific for any case to be analyzed and that it must not be presented as a general model to solve such complex problems as those which live in the healthcare domain.
In the light of this considerations I suggest rejecting the paper but with very strong encouragement to the authors to rework and resubmit after having carefully addressed and managed the SVi9 issues.

**Audience:**

I think that the interest on the subject presented and discussed in the paper could be high and the same view is shared by all three reviewers.

**Claims And Evidence:**

The reviewers reached a mixing picture concerning claims and evidence, indeed, 3L5q and 1vYs answered yes, while SVi9 thinks that claim and evidence are not supported. My opinion is that without a clear theoretical framework, i.e., potential outcomes and/or causal networks, and associated assumptions, any statement and evidence is somewhat weak and thus to support the main claims more discussion is needed.

**Resubmission Of Major Revision:**

The authors may consider submitting a major revision at a later time.

---

> ### Author Response · Authors · 2024-08-16
> **Request for Further Clarification of the Decision (Paper 2730)**
>
> Dear Action Editor 8bgj,
>
>
> We sincerely appreciate your thorough review and constructive feedback on our paper. After receiving this decision, we have immediately started the preparation for the resubmission.
>
>
> To revise our paper in the most proper manner, we want to confirm a few points to grasp the precise meaning of your comments.
>
>
>
> $\mathbf{ Question 1: }$
>
> >I also put particular attention to the review and the argument from the third reviewer who recommended rejection.
>
> In this sentence, we interpret that you call the reviewer SVi9 as the third reviewer, who recommended rejection of our paper. Is it true?
>
> $\mathbf{ Question 2: }$
>
> >If what suggested by SVi9 is achieved can it will significantly improve the impact of the interesting work of the authors, exactly as SVi9 agrees upon in the last comments to the recommendation
>
> We have discussed the meaning of this sentence among all of the authors, and we interpret that "the last comments to the recommendation" means the paragraph below:
>
> >I think this paper itself is worthy of publication. However, current submissions lack rigors in the context of causal discovery. I carefully checked the limitation sections and felt rather limited. I would like this paper could have an effective revision based on
>
> >Clearly illustrate the limitations and risks of using current LLM in data-driven causal discovery.
>
> >Showing the failure scenarios that LLM worsens the causal discovery.
>
> >Clearly showing how the probability of confidence is obtained, including its sensitivity and reliability.
>
> However, as we have stated in the rebuttal, we recognize the suggestions have been already achieved in the revised version uploaded on 17th July.
> Furthermore, we have no way to confirm whether the concerns from reviewer SVi9 have not really been fully addressed, as we have never received additional questions or concerns from reviewer SVi9 at all.
> Thus, we have not found the solution for the satisfaction of this condition.
>
>
> Could you tell us why you have judged that we have not satisfied these suggestions in detail, although we have already incorporated the comments from reviewer SVi9 in our latest manuscript?
>
>
> $\mathbf{ Question 3: }$
>
> >I also think the approach should be made more specific for any case to be analyzed
>
> Does this mean that we should discuss in a clearer way which prompting pattern and with which SCD algorithm is expected to produce the best performance from our experiments?
>
>
> If you really mean so, we believe that this revise has an important meaning, and we think that it is possible to achive this condition only with minor revision.
>
> $\mathbf{ Question 4: }$
>
> >it must not be presented as a general model to solve such complex problems as those which live in the healthcare domain
>
> Does this mean that we should weaken the statements on the detailed possibility of future application, considering the social acceptability in the context of practical causal inference?
>
> If you mean so, we think that there is room to reconsider the expression.  We think that it is possible to achive this condition only with minor revision.
>
>
>
> We value your feedback and are committed to improving our manuscript accordingly.
>
>
>
> Thank you for your time and support.
>
>
> Best regards,
>
> Authors of paper 2730